# LEACE: Perfect linear concept erasure in closed form

**Nora Belrose**[1]    **David Schneider-Joseph**[1]    **Shauli Ravfogel**[2]    **Ryan Cotterell**[3]
**Edward Raff**[4]    **Stella Biderman**[1,4]
[1]EleutherAI    [2]Bar-Ilan University    [3]ETH Zürich    [4]Booz Allen Hamilton
{nora,stella}@eleuther.ai    david@davidsj.com

## Abstract

Concept erasure aims to remove specified features from a representation. It can improve fairness (e.g. preventing a classifier from using gender or race) and interpretability (e.g. removing a concept to observe changes in model behavior). We introduce LEAst-squares Concept Erasure (LEACE), a closed-form method which provably prevents all linear classifiers from detecting a concept while changing the representation as little as possible, as measured by a broad class of norms. We apply LEACE to large language models with a novel procedure called concept scrubbing, which erases target concept information from *every* layer in the network. We demonstrate our method on two tasks: measuring the reliance of language models on part-of-speech information, and reducing gender bias in BERT embeddings. Our code is available at `https://github.com/EleutherAI/concept-erasure`.

## 1  Introduction

The ability to prevent a machine learning system from using a specified concept is important for fairness and interpretability. Popular notions of fairness require that protected attributes should not causally affect predictions [22, 26], and interpretability research often estimates the causal effect of a concept by attempting to remove it from a model's internal representations [10, 30, 25, 5, 18].

What it means for a model $\mathcal{M}$ to "use" a concept Z is often vague and application-specific, but a necessary condition is that its outputs—and therefore its inputs and hidden states—should have significant *mutual information* with Z.[1] **Concept erasure** leverages this fact to limit $\mathcal{M}$'s use of Z *without* finetuning or inspecting its parameters. Instead, we edit the input or hidden states X used by $\mathcal{M}$ to minimize the predictive $\mathcal{V}$-information $I_{\mathcal{V}}(X \to Z)$ [43], a tractable lower bound on the mutual information $I(X; Z)$ which measures the degree to which classifiers from the family $\mathcal{V}$ can predict Z. Intuitively, if no classifier in $\mathcal{V}$ can outperform a constant function at predicting Z—a condition known as **guardedness**—then $\mathcal{M}$ can't use Z either, at least if $\mathcal{V}$ is expressive enough relative to $\mathcal{M}$.

In this work, we improve upon existing concept erasure techniques using a theory-driven approach. We focus on the case where $\mathcal{V}$ is the set of linear classifiers, and prove a previously unnoticed equivalence: a classification task is linearly guarded *if and only if* every class has exactly the same mean feature vector (§ 3). Leveraging this equivalence, we derive a simple necessary and sufficient condition for an affine transformation to produce linearly guarded features. We then identify the unique *surgical* transformation in this family—the one that minimizes the mean squared distance from the original features with respect to *all* norms induced by inner products, including the popular Euclidean and Mahalanobis norms. We name it **LEAst-squares Concept Erasure (LEACE)** (§ 4).

While prior work has focused on preventing linear models from leveraging Z, we aim to erase concepts from deep neural networks as well. Interpretability research has shown that networks

---

[1]This follows from the fact that causal dependence is a special kind of statistical dependence [28]. By the data processing inequality, $\mathcal{M}$'s output can't have any more information about Z than its input or hidden states.

can be usefully described as encoding features in linear subspaces [11, 24, 41], suggesting that fundamentally nonlinear methods may not be necessary for successful erasure in DNNs. In light of this, we introduce a simple procedure called **concept scrubbing** (§ 6), which sequentially applies LEACE to the intermediate representations at each layer of a deep network.

We empirically validate our proposals, demonstrating the superiority of LEACE for erasing gender bias from BERT representations (§ 5.2), and using concept scrubbing to measure the extent to which large language models use part-of-speech information (§ 6).

## 2  Preliminaries

Consider a $k$-class classification task over jointly defined random vectors X (the input data) and Z (the one-hot labels), with X of finite first moment and taking values in $\mathbb{R}^d$, and Z taking values in $\mathcal{Z} = \{\mathbf{z} \in \{0,1\}^k \mid \|\mathbf{z}\|_1 = 1\}^2$ with each $\mathbb{P}(Z = j) > 0$. Let $\eta(\cdot; \boldsymbol{\theta}) : \mathbb{R}^d \to \mathbb{R}^k$ be a predictor chosen from a function class $\mathcal{V} = \{\eta(\cdot; \boldsymbol{\theta}) \mid \boldsymbol{\theta} \in \Theta\}$ (presumed to contain all constant functions) so as to minimize the expectation $\mathbb{E}\big[\mathcal{L}(\eta(X), Z)\big]$ of some $\mathcal{L} : \mathbb{R}^k \times \mathcal{Z} \to [0, \infty)$ in a class $\mathfrak{L}$ of loss functions.

We borrow the concept of **guardedness** from Ravfogel et al. [31], who define it in terms of $\mathcal{V}$-information [43]. We opt for a slightly more general definition here, which is equivalent to theirs in the case of cross-entropy loss (see Appendix G).

**Definition 2.1** (Guardedness). *Let* X, Z, $\mathcal{V}$, *and* $\mathfrak{L}$ *be as defined above, and let* $\chi$ *be the set of all random vectors of finite first moment taking values in* $\mathbb{R}^d$, *jointly defined with* Z.

*We say* X $(\mathcal{V}, \mathfrak{L})-$**guards** Z *if, for all losses* $\mathcal{L} \in \mathfrak{L}$, *it maximizes the minimum expected loss:*

$$\text{X} \in \underset{\text{X}' \in \chi}{\operatorname{argmax}} \ \underset{\boldsymbol{\theta} \in \Theta}{\inf} \ \mathbb{E}\Big[\mathcal{L}(\eta(\text{X}'; \boldsymbol{\theta}), Z)\Big].$$

*In other words, its conditional distribution* $\mathbb{P}(\text{X} \mid Z = \cdot)$ *is among the worst possible distributions for predicting* Z *from* X *using a predictor of the form* $\eta(\cdot; \boldsymbol{\theta}) \in \mathcal{V}$ *and a loss function in* $\mathfrak{L}$.

**Definition 2.2** (Trivially Attainable Loss). *The **trivially attainable loss** for labels* Z *and loss* $\mathcal{L}$ *is the lowest possible expected loss available to a constant predictor* $\eta(\mathbf{x}) = \mathbf{b}$:

$$L_\tau = \underset{\mathbf{b} \in \mathbb{R}^k}{\inf} \ \mathbb{E}[\mathcal{L}(\mathbf{b}, Z)]$$

*We will sometimes write it* $L_\tau^{(Z, \mathcal{L})}$ *in cases of possible ambiguity. If there is a specific constant predictor actually achieving this loss, we call it the **trivial predictor*** $\eta_\tau = \eta_\tau^{(Z, \mathcal{L})}$.

We examine this problem in the important case of loss functions $\mathcal{L} : \mathbb{R}^k \times \mathcal{Z} \to [0, \infty)$ which are convex in the prediction $\eta(\mathbf{x})$, and linear predictors that take the functional form $\eta(\mathbf{x}; \mathbf{b}, \mathbf{W}) = \mathbf{b} + \mathbf{W}\mathbf{x}$, for some bias $\mathbf{b} \in \mathbb{R}^k$ and weight matrix $\mathbf{W} \in \mathbb{R}^{k \times d}$.

**Definition 2.3** (Linear Guardedness). *If* X $(\mathcal{V}, \mathfrak{L})$-*guards* Z, *where* $\mathfrak{L}$ *is the class of nonnegative loss functions which are convex in their first argument, and* $\mathcal{V}$ *is the class of linear predictors* $\eta(\mathbf{x}) = \mathbf{b} + \mathbf{W}\mathbf{x}$, *we say that* X **linearly guards** Z.

## 3  Theoretical Results

Our primary theoretical result is that the following conditions are all equivalent:

1. The data X linearly guards the labels Z. (Definition 2.3)
2. For all convex losses $\mathcal{L}$, the trivially attainable loss is optimal on $(X, Z)$. (Definition 2.2)
3. The class-conditional mean vectors $\mathbb{E}[X \mid Z = i]$ are equal to the unconditional mean $\mathbb{E}[X]$.
4. Every component of X has zero covariance with every component of Z.
5. Every linear classifier evaluated on X exhibits statistical parity w.r.t. Z. (App. C)

The equivalence of conditions 1, 2, and 5 is relatively straightforward to show, and the relevant theorems can be found in Appendices B and C. The other equivalences are proven below (cond. 3 ↔ cond. 2 in § 3.1 and § 3.2); cond. 3 ↔ 4 in § 3.3).

---

[2] We frequently use the integer $j \le k$ to refer to the element of $\mathcal{Z}$ which is 1 at the $j^{\text{th}}$ index and 0 elsewhere.

## 3.1 Equality of Class Centroids Implies Linear Guardedness

The following result establishes the implication from condition 3 to condition 2.

**Theorem 3.1.** *Suppose $\mathcal{L}$ is convex in the linear prediction $\eta$. Then if each class-conditional mean $\mathbb{E}\big[\mathrm{X} \mid \mathrm{Z} = i\big]$ is equal to $\mathbb{E}\big[\mathrm{X}\big]$, the trivially attainable loss cannot be improved upon.*

*Proof.* Let $\eta(\mathbf{x}) = \mathbf{b} + \mathbf{W}\mathbf{x}$ be any linear predictor. By Jensen's inequality,[3] the loss with $\eta$ evaluated on X is lower bounded by the loss with $\eta$ evaluated on the unconditional mean of the data $\mathbb{E}\big[\mathrm{X}\big]$:

$$
\begin{aligned}
\mathbb{E}\Big[\mathcal{L}(\eta, \mathrm{Z})\Big] &= \mathbb{E}_{\mathrm{Z}}\Big[\mathbb{E}\big[\mathcal{L}(\eta, \mathrm{Z})|\mathrm{Z}\big]\Big] \\
&\geq \mathbb{E}_{\mathrm{Z}}\Big[\mathcal{L}\big(\mathbb{E}[\eta|\mathrm{Z}], \mathrm{Z}\big)\Big] && \text{(Jensen's inequality)} \\
&= \mathbb{E}_{\mathrm{Z}}\Big[\mathcal{L}\big(\mathbf{b} + \mathbf{W}\mathbb{E}\big[\mathrm{X}|\mathrm{Z}\big], \mathrm{Z}\big)\Big] && \text{(linearity of } \eta) \\
&= \mathbb{E}_{\mathrm{Z}}\Big[\mathcal{L}\big(\mathbf{b} + \mathbf{W}\mathbb{E}\big[\mathrm{X}\big], \mathrm{Z}\big)\Big]. && \text{(by assumption)}
\end{aligned}
$$

This in turn is the loss of the constant predictor $\eta'(\mathbf{x}) = \mathbf{b} + \mathbf{W}\mathbb{E}\big[\mathrm{X}\big]$. Since the trivially attainable loss is the best that can be achieved by a constant predictor, and *every* predictor's loss is lower bounded by that of some constant predictor, we cannot improve upon the trivially attainable loss. □

Intuitively, this shows that the classifier's expected loss is lower-bounded by the loss it would receive if each data point were replaced with the centroid of its class. But, if these centroids are all equal, the loss can't be any lower than what we'd get if every data point were replaced with the *global* mean $\mathbb{E}[\mathrm{X}]$. In that case, the data points are indistinguishable and we can't do better than $\mathbf{W} = \mathbf{0}$.

## 3.2 Linear Guardedness Implies Equality of Class Centroids

We now prove the implication from condition 2 to condition 3. Condition 2 applies when the trivially attainable loss is optimal for *all* convex losses, including cross-entropy loss in particular. And if it holds for cross-entropy loss, we now show that condition 3—the class centroids are equal—must follow. First a more general lemma:

**Lemma 3.2.** *Suppose $\mathcal{L}$ has bounded partial derivatives, which when off-category never vanish and do not depend on the category, i.e. $\partial\mathcal{L}(\eta, z_1)/\partial\eta_i = \partial\mathcal{L}(\eta, z_2)/\partial\eta_i \neq 0$ for all categories $z_1, z_2 \neq i$. If $\mathbb{E}\big[\mathcal{L}(\eta, \mathrm{Z})\big]$ is minimized among linear predictors by the constant predictor $\eta(\mathbf{x}) = \mathbf{b}^* + \mathbf{W}^*\mathbf{x}$ with $\mathbf{W}^* = \mathbf{0}$, then each class-conditional mean $\mathbb{E}\big[\mathrm{X}|\mathrm{Z} = i\big]$ is equal to $\mathbb{E}\big[\mathrm{X}\big]$.*

*Proof.* The first-order optimality condition on the $i^{\text{th}}$ component of our parameters $\mathbf{b}$ and $\mathbf{W}$ yields the equations:

$$
\mathbb{E}\left[\frac{\partial\mathcal{L}(\eta, \mathrm{Z})}{\partial\eta_i} \cdot \frac{\partial\eta_i}{\partial b_i}\right] = 0 \quad \text{and} \quad \mathbb{E}\left[\frac{\partial\mathcal{L}(\eta, \mathrm{Z})}{\partial\eta_i} \cdot \frac{\partial\eta_i}{\partial\mathbf{W_i}}\right] = \mathbf{0}, \tag{1}
$$

where we have used the boundedness of $\mathcal{L}$'s partial derivative and the finite first moment of $\frac{\partial\eta_i}{\partial b_i} = 1$ and $\frac{\partial\eta_i}{\partial\mathbf{W_i}} = \mathrm{X}$ to justify (via the Dominated Convergence Theorem) interchanging the derivative with the expectation.

Since $\eta$ is constant over all values of X, and $\frac{\partial\eta_i}{\partial b_i} = 1$, the first equation in (1) reduces to:

$$
\mathbb{P}\big(\mathrm{Z} = i\big)\frac{\partial\mathcal{L}(\eta, i)}{\partial\eta_i} + \mathbb{P}(\mathrm{Z} \neq i)\frac{\partial\mathcal{L}(\eta, \neq i)}{\partial\eta_i} = 0, \tag{2}
$$

where $\frac{\partial\mathcal{L}(\eta, \neq i)}{\partial\eta_i}$ is an abuse of notation denoting the off-category partial derivative, emphasizing its independence of the category Z.

---

[3] Specifically, its generalization to convex functions over $\mathbb{R}^k$. See [12] p. 76.

Similarly, the constancy of $\eta$ and the fact that $\frac{\partial \eta_i}{\partial \mathbf{W_i}} = X$ reduces the second equation in (1) to:

$$\mathbb{P}(Z = i)\frac{\partial \mathcal{L}(\eta, i)}{\partial \eta_i} \cdot \mathbb{E}\big[X|Z = i\big] + \mathbb{P}(Z \neq i)\frac{\partial \mathcal{L}(\eta, \neq i)}{\partial \eta_i} \cdot \mathbb{E}\big[X|Z \neq i\big] = \mathbf{0}. \qquad (3)$$

Solving for $\mathbb{P}(Z = i)\frac{\partial \mathcal{L}(\eta, i)}{\partial \eta_i}$ in (2) and substituting in (3) gives us:

$$\mathbb{P}(Z \neq i)\frac{\partial \mathcal{L}(\eta, \neq i)}{\partial \eta_i} \cdot \left( \mathbb{E}\big[X|Z \neq i\big] - \mathbb{E}\big[X|Z = i\big] \right) = \mathbf{0}.$$

If $\mathbb{P}(Z \neq i) = 0$, then $\mathbb{E}[X] = \mathbb{E}[X|Z = i]$ is trivially true. Otherwise, using the non-vanishingness of the off-category partial derivative $\frac{\partial \mathcal{L}(\eta, \neq i)}{\partial \eta_i}$, division yields the equivalence of $\mathbb{E}\big[X|Z = i\big]$ to $\mathbb{E}\big[X|Z \neq i\big]$, and hence to the unconditional mean $\mathbb{E}\big[X\big]$. $\qquad \square$

We now show that Lemma 3.2 applies to the widely used cross entropy loss:

**Theorem 3.3.** *If the class probabilities $\mathbb{P}(Z = j)$ are all nonzero, and the trivially obtainable loss is optimal when $\mathcal{L}(\eta, z) = -\log \frac{\exp(\eta_z)}{\sum_{i=1}^{k} \exp(\eta_i)}$, then each class has the same mean $\mathbb{E}\big[X|Z = z\big]$.*

*Proof.* In this case, the trivial predictor $\eta_\tau(Z)_j = \log(\mathbb{P}(Z = j))$ exists, achieving the trivially obtainable loss, which we have assumed optimal. Furthermore, $\mathcal{L}$ has on-category partial derivative $\partial \mathcal{L}(\eta, i)/\partial \eta_i = \exp(\eta_i)/\sum_{j=1}^{k} \exp(\eta_j) - 1 \in (-1, 0]$, and nonvanishing off-category partial derivative $\partial \mathcal{L}(\eta, \neq i)/\partial \eta_i = \exp(\eta_i)/\sum_{j=1}^{k} \exp(\eta_j) \in (0, 1)$, both bounded, so the conditions of Lemma 3.2 apply. $\qquad \square$

### 3.3 Linearly Guarded Labels Have Zero Covariance with the Features

The next theorem establishes the equivalence of conditions 3 and 4.

**Theorem 3.4.** *Let $X$ be a random vector taking values in $\mathbb{R}^d$ with finite first moment, and $Z$ a random vector taking values in $\{0, 1\}^k$ with one-hot encoding, with each class probability $\mathbb{P}(Z = j)$ being nonzero. Then the class-conditional means $\mathbb{E}[X|Z = j]$ are all equal to the unconditional mean $\mathbb{E}[X]$ if and only if every component of $X$ has zero covariance with every component of $Z$, i.e. the cross-covariance matrix $\mathbf{\Sigma}_{XZ}$, whose $(i, j)^{th}$ entry is $\mathrm{Cov}(X_i, Z_j)$, is the zero matrix.*

*Proof.* Since $Z$ is one-hot, we can rewrite the $(i, j)^{\text{th}}$ entry of $\mathbf{\Sigma}_{XZ}$ as:

$$\mathbb{E}[X_i Z_j] - \mathbb{E}[X_i]\mathbb{E}[Z_j] = \mathbb{P}(Z = j)\Big( \mathbb{E}[X_i|Z = j] - \mathbb{E}[X_i] \Big).$$

As $\mathbb{P}(Z = j) > 0$, it follows that $\mathbb{E}[X_i|Z = j] = \mathbb{E}[X_i]$ if and only if $\mathrm{Cov}(X_i, Z_j) = 0$. $\qquad \square$

We have thus established the equivalence of the first four conditions stated earlier. See Appendix C for the last one, on statistical parity.

## 4 Least-Squares Concept Erasure

In Section 3 we saw that $X$ linearly guards $Z$ if and only if each component of $X$ has zero covariance with each component of $Z$. We will now characterize the set of affine transformations $r(\mathbf{x}) = \mathbf{P}\boldsymbol{x} + \mathbf{b}$ such that $r(X)$ linearly guards $Z$.

**Theorem 4.1.** *Let $X$ and $Z$ be random vectors taking values in $\mathbb{R}^d$ and $\mathbb{R}^k$ respectively, with $X$ of finite first moment. Then given some affine function $r(\boldsymbol{x}) = \mathbf{P}\boldsymbol{x} + \mathbf{b}$, the modified random vector $r(X)$ linearly guards $Z$ if and only if the columns of the cross-covariance matrix $\mathbf{\Sigma}_{XZ}$ are contained in the null space of $\mathbf{P}$.*

*Proof.* From Theorem 3.4 we know that $r(X)$ linearly guards $Z$ if and only if $\mathrm{Cov}(r(X), Z)$ is the zero matrix. By the linearity property of cross-covariance, we have:

$$\mathrm{Cov}(r(X), Z) = \mathrm{Cov}(\mathbf{P}X + \mathbf{b}, Z) = \mathbf{P}\mathrm{Cov}(X, Z) = \mathbf{P}\mathbf{\Sigma}_{XZ}.$$

Therefore, $r(X)$ linearly guards $Z$ if and only if $\ker(\mathbf{P}) \supseteq \mathrm{colsp}(\mathbf{\Sigma}_{XZ})$. $\qquad \square$

**Implications for prior work.** Notably, the above theorems imply that three previously proposed methods in the literature, Spectral Attribute Removal (SAL) [37], Mean Projection [17], and Fair PCA [20], are guaranteed to achieve linear guardedness given suitable hyperparameters. See Appendix D for further discussion.

## 4.1 Derivation of LEACE

Theorem 4.1 is a very weak condition, which is far from identifying unique values for $\mathbf{P}$ and $\mathbf{b}$. In most applications, however, we'd like to make a "small" edit to X so that useful information contained in X is maximally preserved. We operationalize the notion of a small edit in terms of the mean squared norm $\mathbb{E}\|r(\mathrm{X}) - \mathrm{X}\|_{\mathbf{M}}^2$ defined by some positive-definite inner product $\mathbf{M}$,[4] which can be thought of as a local quadratic approximation to *any* measure of divergence between X and $r(\mathrm{X})$ (such as Kullback–Leibler divergence, for example). While we are primarily interested in the Euclidean ($\mathbf{M} = \mathbf{I}$) and Mahalanobis ($\mathbf{M} = \mathbf{\Sigma}_{\mathrm{XX}}^+$) norms, it will turn out that there is a *single* erasure function that minimizes *all* such norms simultaneously. We will see in Section 6 that ensuring edits are small in this sense provides substantial benefit to downstream task performance as compared to other methods which also guard the labels Z.

Below, we derive the optimal eraser under the assumption that X and Z are centered.

**Theorem 4.2.** *Let* X *and* Z *be centered random vectors taking values in $\mathbb{R}^d$ and $\mathbb{R}^k$ respectively, each of finite second moment. Let $\mathbf{M} \in \mathbb{R}^{d \times d}$ be a p.s.d. matrix defining a (possibly degenerate) inner product on $\mathbb{R}^d$: $\langle \mathbf{x}, \mathbf{y} \rangle_{\mathbf{M}} = \mathbf{x}^T \mathbf{M} \mathbf{y}$. Let $\mathbf{\Sigma}_{\mathrm{XX}} \in \mathbb{R}^{d \times d}$ be X's covariance matrix, and $\mathbf{\Sigma}_{\mathrm{XZ}} \in \mathbb{R}^{d \times k}$ be the cross-covariance matrix of* X *and* Z. *Let $\mathbf{A}^+$ denote the Moore-Penrose pseudoinverse of a matrix $\mathbf{A}$, and let $\mathbf{A}^{1/2}$ be the p.s.d. square root of a p.s.d. matrix $\mathbf{A}$. Then the objective*

$$\operatorname*{argmin}_{\mathbf{P} \in \mathbb{R}^{d \times d}} \mathbb{E}\left[\left\|\mathbf{P}\mathrm{X} - \mathrm{X}\right\|_{\mathbf{M}}^2\right] \quad \text{subject to } \operatorname{Cov}(\mathbf{P}\mathrm{X}, \mathrm{Z}) = \mathbf{0}$$

*has the following solution:*

$$\mathbf{P}^* = \mathbf{I} - \mathbf{W}^+ \mathbf{P}_{\mathbf{W}\mathbf{\Sigma}_{\mathrm{XZ}}} \mathbf{W},$$

*where $\mathbf{W}$ is the whitening transformation $(\mathbf{\Sigma}_{\mathrm{XX}}^{1/2})^+$ and $\mathbf{P}_{\mathbf{W}\mathbf{\Sigma}_{\mathrm{XZ}}} = (\mathbf{W}\mathbf{\Sigma}_{\mathrm{XZ}})(\mathbf{W}\mathbf{\Sigma}_{\mathrm{XZ}})^+$ is the orthogonal projection matrix onto $\operatorname{colsp}(\mathbf{W}\mathbf{\Sigma}_{\mathrm{XZ}})$.*

*Proof.* See Appendices E.1 and E.2 for two independent proofs of Theorem 4.2. □

The above theorem assumes that the random vectors X and Z are centered, and does not include a bias term. Below we extend our results to the uncentered case, and derive the optimal bias $\mathbf{b}^*$.

**Theorem 4.3.** *Let* X *and* Z *be random vectors taking values in $\mathbb{R}^d$ and $\mathbb{R}^k$ respectively, each of finite second moment. Define $\mathbf{M}$ and $\mathbf{P}^*$ as in Theorem 4.2 and $\mathbf{b}^* = \mathbb{E}[\mathrm{X}] - \mathbf{P}^*\mathbb{E}[\mathrm{X}]$. Then $(\mathbf{P}^*, \mathbf{b}^*)$ minimizes $\mathbb{E}\|\mathbf{P}\mathrm{X} + \mathbf{b} - \mathrm{X}\|^2$, subject to $\operatorname{Cov}(\mathbf{P}\mathrm{X} + \mathbf{b}, \mathrm{Z}) = \mathbf{0}$.*

*Proof.* Let $\mathbf{P} \in \mathbb{R}^{d \times d}$ and define $\tilde{\mathrm{X}} = \mathrm{X} - \mathbb{E}[\mathrm{X}]$ and $\mathbf{c} = \mathbf{P}\mathbb{E}[\mathrm{X}] + \mathbf{b} - \mathbb{E}[\mathrm{X}]$. Then,

$$\mathbb{E}\|\mathbf{P}\mathrm{X} + \mathbf{b} - \mathrm{X}\|_{\mathbf{M}}^2 = \mathbb{E}\|(\mathbf{P}\tilde{\mathrm{X}} - \tilde{\mathrm{X}}) + \mathbf{c}\|_{\mathbf{M}}^2$$

$$= \mathbb{E}\|\mathbf{P}\tilde{\mathrm{X}} - \tilde{\mathrm{X}}\|_{\mathbf{M}}^2 + 2\mathbb{E}\big[\mathbf{P}\tilde{\mathrm{X}} - \tilde{\mathrm{X}}\big]^T \mathbf{M}\mathbf{c} + \mathbf{c}^T \mathbf{M}\mathbf{c}$$

$$= \mathbb{E}\|\mathbf{P}\tilde{\mathrm{X}} - \tilde{\mathrm{X}}\|_{\mathbf{M}}^2 + \mathbf{c}^T \mathbf{M}\mathbf{c},$$

where we have eliminated the middle term because $\mathbf{P}$ is linear and $\mathbb{E}[\tilde{\mathrm{X}}] = 0$. Since $\mathbf{M}$ is p.s.d., our objective is minimized for $\mathbf{c} = \mathbf{0}$, i.e. $\mathbf{b} = \mathbb{E}[\mathrm{X}] - \mathbf{P}\mathbb{E}[\mathrm{X}]$. The problem thus reduces to choosing $\mathbf{P}$ so as to minimize $\mathbb{E}\|\mathbf{P}\tilde{\mathrm{X}} - \tilde{\mathrm{X}}\|_{\mathbf{M}}^2$ subject to $\operatorname{Cov}(\mathbf{P}\mathrm{X} + \mathbf{b}, \mathrm{Z}) = \operatorname{Cov}(\mathbf{P}\tilde{\mathrm{X}}, \mathrm{Z}) = \mathbf{0}$, which Theorem 4.2 shows occurs when $\mathbf{P} = \mathbf{P}^*$. □

---

[4]Our proofs also include degenerate "inner products" where $\mathbf{M}$ is singular, and the associated seminorms.

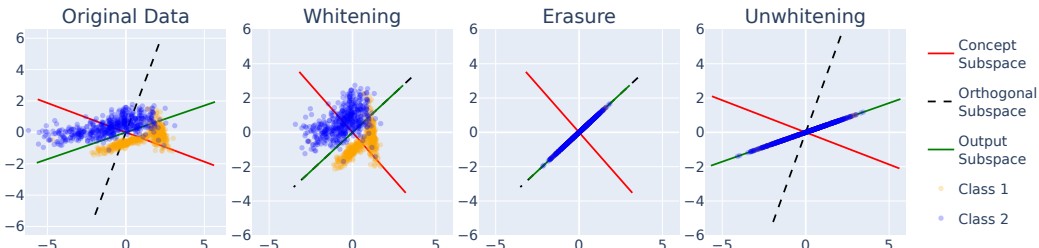

Figure 1: LEACE projection in 3 steps. First the data is whitened, ensuring equal variance in all directions. It is then orthogonally projected onto $\text{colsp}(\mathbf{W}\boldsymbol{\Sigma}_{\text{XZ}})^{\perp}$, guaranteeing linear guardedness. Finally, we unwhiten the data so that its covariance structure mimics the original.

Putting together Theorems 4.2 and 4.3 and rearranging, we arrive at the LEACE formula:

$$r_{\text{LEACE}}(\boldsymbol{x}) = \boldsymbol{x} - \mathbf{W}^{+}\mathbf{P}_{\mathbf{W}\boldsymbol{\Sigma}_{\text{XZ}}}\mathbf{W}\big(\boldsymbol{x} - \mathbb{E}[\text{X}]\big) \qquad (1)$$

Intuitively, LEACE de-means and whitens $\boldsymbol{x}$, projects onto the subspace responsible for correlations between X and Z, then unwhitens the result. Finally, it subtracts this value from $\boldsymbol{x}$, thereby surgically removing the linearly available information about Z.

## 4.2 Oblique Projections are Least-Squares Optimal

Prior work on linear concept erasure has assumed that erasure functions should be orthogonal projections [29, 33, 37], appealing to the well-known fact that an orthogonal projection of a point $\boldsymbol{x}$ onto a subspace $U$ yields the nearest point in $U$ to $\boldsymbol{x}$. But even in the case where X is centered, $r_{\text{LEACE}}$ is *not* an orthogonal projection in general. Orthogonal projection matrices are symmetric, and $\mathbf{I} - \mathbf{W}^{+}\mathbf{P}_{\mathbf{W}\boldsymbol{\Sigma}_{\text{XZ}}}\mathbf{W}$ is only symmetric in the special case where $\mathbf{P}_{\mathbf{W}\boldsymbol{\Sigma}_{\text{XZ}}}$ and $\mathbf{W}$ commute. It is an *oblique* projection however, since applying $\mathbf{P}^{*}$ twice yields the same result as applying it once: $(\mathbf{P}^{*})^{2} = \mathbf{I} - 2\mathbf{W}\mathbf{P}_{\mathbf{W}\boldsymbol{\Sigma}_{\text{XZ}}}\mathbf{W}^{+} + \mathbf{W}^{+}\mathbf{P}_{\mathbf{W}\boldsymbol{\Sigma}_{\text{XZ}}}\mathbf{W}\mathbf{W}^{+}\mathbf{P}_{\mathbf{W}\boldsymbol{\Sigma}_{\text{XZ}}}\mathbf{W} = \mathbf{P}^{*}$.

Orthogonal projections are generally not least-squares optimal for concept erasure because the necessary and sufficient condition for linear guardedness, $\mathbf{P}\boldsymbol{\Sigma}_{\text{XZ}} = \mathbf{0}$, is a constraint on the *nullspace* of $\mathbf{P}$, and not on its range. We may freely choose the range of the projection to minimize the mean squared distance, as long as we zero out $\text{colsp}(\boldsymbol{\Sigma}_{\text{XZ}})$. In Figure 1, an orthogonal projection would map all points onto the the dashed line, thereby preserving less of the variance of the original data than LEACE does (green line). See Appendix F for a concrete example.

## 4.3 Extension to Continuous Z

While not a focus of this work, it's worth noting that LEACE can also be applied to the setting where Z takes arbitrary values in $\mathbb{R}^{k}$, as long as we restrict ourselves to the ordinary least squares regression loss $\mathcal{L}(\eta, \mathbf{z}) = \|\eta - \mathbf{z}\|_{2}^{2}$. In particular, the proofs of equivalence between conditions 1 and 2 given in Appendix B make no categorical assumption on Z, and the equivalence between the optimality of a zero weight matrix (condition 2) and zero cross-covariance (condition 4) is well known in the OLS setting. We can then apply Theorems 4.2 and 4.3, which also make no categorical assumption, to derive the same optimal affine eraser as in the categorical case.

# 5 Evaluation

## 5.1 Intrinsic Evaluation

Following Ravfogel et al. [32] we evaluate the ability of our method to remove gender information from the last hidden layer of a frozen BERT model. We use the biographies dataset of De-Arteaga et al. [6], composed of short biographies annotated by both binary gender and profession. We embed each biography with the [CLS] representation in the last layer of BERT, enforce the same-conditional-mean constraint to remove gender information from the [CLS], and then evaluate the performance of the model, after the intervention, on the main task of profession prediction. We

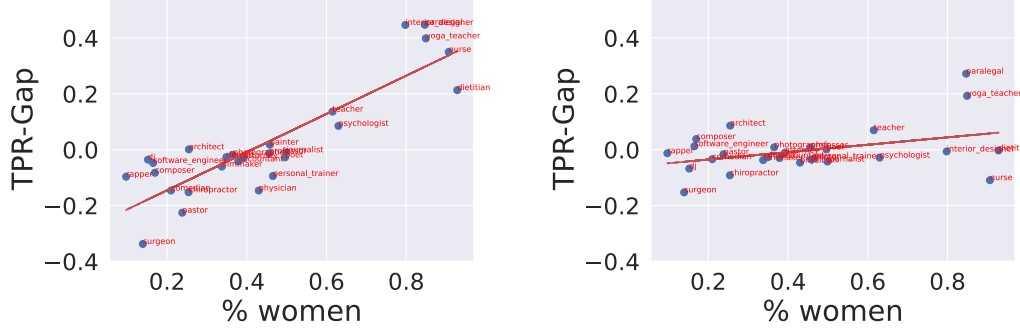

Figure 3: The correlation between $GAP^{TPR}_{female,y}$ and the relative proportion of women in profession $y$, for BERT representation, before (left; R=0.867) and after (right; R=0.392) the projection.

compare our intervention with RLACE [32], which uses gradient-based optimization to solve a linear concept-erasure adversarial game.

**Concept erasure results.** First, we evaluate the ability of logistic regression classifiers to recover the removed information. The results, presented in Fig. 2, show that our method is the only to achieve random accuracy (perfect erasure) with a small edit, although RLACE (but not INLP) comes close. At the same time, our method is around 2 orders of magnitude faster, and does not require gradient-based optimization.

## 5.2 Downstream Fairness

How does our intervention affect the behavior of the model on the main classification task of profession prediction? We fit a logistic regression profession-prediction classifier over the projected [CLS] representations.

To measure the bias in a classifier, we follow De-Arteaga et al. [6] and use the TPR-GAP measure, which quantifies the bias in a classifier by considering the difference (GAP) in the true positive rate (TPR) between individuals with different protected attributes (e.g. race or gender). We use the notation $GAP^{TPR}_{z,y}$ to denote the TPR-gap in some main-class label $y$ (e.g. "nurse" prediction) for some protected group $z$ (e.g. "female"), we also consider $GAP^{TPR,RMS}_z$, the RMS of the TPR-gap across all professions for a protected group $z$:

$$GAP^{TPR,RMS}_z = \sqrt{\frac{1}{|C|}\sum_{y \in C}(GAP^{TPR}_{z,y})^2}$$

To calculate the relation between the bias the model exhibits and the bias in the data, we also calculate $\sigma_{(GAP^{TPR},\%Women)}$, the correlation between the TPR gap in a given profession and the percentage of women in that profession.

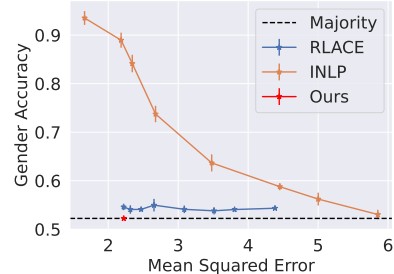

Figure 2: Gender prediction accuracy after bias-removal projection versus the mean squared distance from the original representation for INLP, RLACE, and LEACE on BERT representations.

**Results.** The main-task classifier achieves profession-prediction accuracy of 77.3% on the projected representations (compared with 79.3% over the original representations), indicating that the intervention minimally affects the ability to predict the profession of a person from the representation of their biography. At the same time, the TPR gap drops significantly from 0.198 to 0.084, indicating a sharp drop in the biased behavior of the profession classifier. Indeed, inspecting the correlation $\sigma_{(GAP^{TPR},\%Women)}$ between the gap (per profession) and the representation of women in

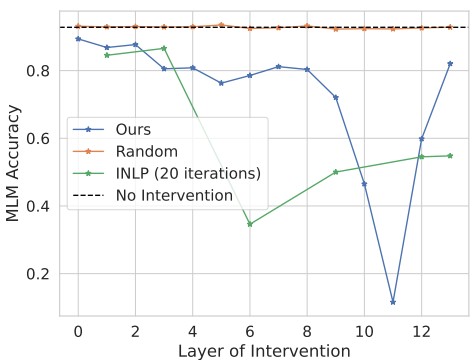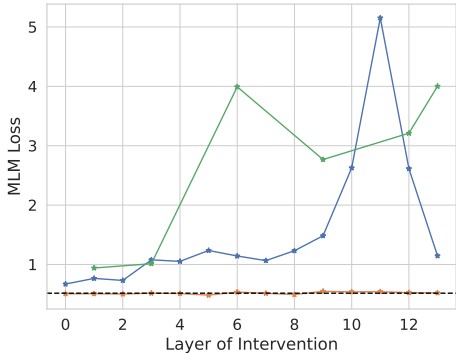

Figure 4: Amnesic probing results on `bert-base-uncased`.

this profession, we see that this correlation plummets from 0.867 to 0.392 after erasure. Re-fitting the main-task logistic regression classifier over the projected representations yields a slightly higher main-task accuracy of 78.1%, at the price of significantly increasing the TPR gap to 0.158.[5]

## 5.3 Revisiting Amnesic Probing

Elazar et al. [10] have introduced the idea of *amnesic probing* as a causal intervention that aims to test the importance of a given concept (e.g. part-of-speech tag) to some main task (e.g. language modeling). They applied Iterative Nullspace Projection (INLP) to remove different concepts from the hidden representations of the model, and assessed the degree to which its behavior changed when performing masked language modeling. Since INLP often requires dozens of iterations to completely erase the concept, its usage in this context raises concerns of collateral damage due to magnitude of the intervention and the non-exhaustive nature of INLP removal. Here, we replicate their experiments on the `bert-base-uncased` model with our interventions.

**Experimental setup.** We use part-of-speech (POS) tags as our concept of interest. We collect sentences and their coarse POS tags ("Noun", "Verb" etc.; 18 in total) from the English Universal Dependencies dataset [27]. We tokenize the sentences with the `BERT` tokenizer and map each word-piece to the POS tag of the word to which it belongs. We collect the unmasked `BERT` representations for each layer, intervene to linearly erase the POS concept from that layer, and continue the forward pass until the last layer, from which we compute the distribution of the MLM over the vocabulary. Note that in each experiment we intervene on a single layer. We quantify the decrease in accuracy following the intervention, as well as the increase in the loss. We compare with a baseline intervention of a random orthogonal projection whose null space has the same rank as the label space (18). For INLP, we perform 20 iterations. This is needed because INLP does not effectively remove the concept; even after 20 iterations, classification accuracy is above majority accuracy. As a result, INLP reduces the rank of the representation by 360. By contrast, our method decreases the rank just by 17.

**Results.** The results are shown in Fig. 4b. Our intervention only mildly changes BERT LM accuracy and loss until layer 8, with the highest drop recorded in layer 11. INLP, in contrast, shows maximum effect at layer 6. Since it removes hundreds of dimensions, it is difficult to attribute this effect to the erasure of the concept. These results suggest that the *causal* effect of the POS concept on the language model is concentrated in layer 11. Interestingly, this stands in contrast with POS linear probing results, which are optimal at earlier layers [38]. As Elazar et al. [10] have noted, probing does not generally correlate with intervention-based analysis techniques.

---

[5]The softmax probabilities of a multiclass logistic regression classifier can leak the removed information if *another* classifier is stacked on top of it [31], though this setup is not linear.

# 6 Concept Scrubbing

Unfortunately, Elazar et al. [10] were forced to limit their interventions to a single layer due to the limitations of INLP. INLP often requires the deletion of several dozen dimensions before linear guarding is achieved—as demonstrated in Figure 2. Kumar et al. [21] show empirically and theoretically that INLP causes needless "collateral damage" to useful parts of the representation that are orthogonal to the concept being erased. Because of this collateral damage, it's impossible to apply INLP to multiple layers of a transformer without causing its outputs to collapse into gibberish.

---

**Algorithm 1** Concept scrubbing

**Require:** Model with $\ell$ layers $f = f_\ell \circ \ldots \circ f_1$
**Require:** Design matrix $\mathbf{X} \in \mathbb{R}^{n \times d}$
**Require:** Label matrix $\mathbf{Z} \in \mathbb{R}^{n \times k}$
**Ensure:** LEACE parameters for each layer in $f$
1: $\mathbf{H}_1 \leftarrow \text{Embed}(\mathbf{X})$
2: $L \leftarrow \texttt{list()}$
3: **for** $l \in 1 \ldots \ell$ **do**
4:    Fit $(\mathbf{P}, \mathbf{b})$ on $\mathbf{H}_l$ and $\mathbf{Z}$
5:    Append $(\mathbf{P}, \mathbf{b})$ to $L$
6:    $\mathbf{H}_l \leftarrow \mathbf{P}(\mathbf{H}_l - \mu_{\mathbf{H}_l}) + \mu_{\mathbf{H}_l}$   (Eq. 1)
7:    $\mathbf{H}_{l+1} \leftarrow f_l(\mathbf{H}_l)$
8: **return** $L$

---

Instead, we would like to erase all linear information about a concept in *every* intermediate representation, which we term **concept scrubbing**. LEACE makes concept scrubbing possible and eminently practical. It causes minimal collateral damage, induces little computational overhead, and the covariance statistics it relies on can be computed in a *streaming* fashion, without ever storing all the hidden states in memory or on disk.

**Algorithm.** Any intervention on the model at layer $\ell$ changes the distribution of hidden states at layers $\ell' > \ell$. Because of this, the naive approach of independently fitting LEACE parameters $(\mathbf{P}, \mathbf{b})$ for all layers of the clean model, then applying them all at once, may fail to fully erase the target concept. Instead, we fit LEACE parameters *sequentially*, starting from the first layer and proceeding to the final layer. After we compute $(\mathbf{P}, \mathbf{b})$ for a layer, we immediately use them to scrub the hidden states for that layer, then feed these scrubbed representations to the next layer (Algorithm 1).

| | LLaMA | | | Pythia | | | |
| Condition | 7B | 13B | 30B | 160M | 1.4B | 6.9B | 12B |
|---|---|---|---|---|---|---|---|
| No intervention | 0.69 | 0.66 | 0.62 | 0.90 | 0.70 | 0.64 | 0.62 |
| Random erasure | 0.69 | 0.66 | 0.62 | 0.99 | 0.72 | 0.66 | 0.63 |
| LEACE | 1.73 | 1.84 | 1.96 | 2.79 | 2.25 | 3.57 | 3.20 |
| SAL | 3.24 | 3.26 | 3.16 | 3.53 | 3.44 | 4.17 | 4.69 |
| unigram entropy | 2.90 | 2.90 | 2.90 | 2.66 | 2.66 | 2.66 | 2.66 |

Table 1: Perplexity in autoregressive language models when removing linearly available part-of-speech information from the input to each transformer layer. Units are bits per UTF-8 byte. The unigram baseline assigns probabilities to tokens based only on their frequency and not on the context.

## 6.1 Experimental Details

**Dataset.** For each model family, we use a sample from the respective pretraining distribution: the validation split of the Pile [13] for the Pythia models [2], and the RedPajama replication of the LLaMA pretraining corpus for the LLaMA family [39]. sampling a slice of $2^{22}$ tokens for fitting the LEACE parameters and another slice of $2^{22}$ tokens for evaluation. Since neither corpus comes with part-of-speech tags, we use the model from the SpaCy library [19] to automatically generate Universal Dependency tags [23].

**Baseline method.** We also run concept scrubbing using full-rank SAL [37], which is similar to our method but lacks a bias term and does not adjust for correlations between features (Appendix D).

**Architecture.** We focus on autoregressive language models. We evaluate our method on EleutherAI's Pythia 160M, 1.4B, 6.9B, and 12B models [2], and Meta's LLaMA 7B, 13B, and 30B [39]. We apply concept erasure to the input of each transformer block, immediately after normalization is applied (LayerNorm or RMSNorm).

**Randomized erasure.** Almost any intervention on a neural network will cause its performance to degrade to some extent. Following Elazar et al. [10], we isolate the effect of the concept erasure by comparing it to a control condition in which we orthogonally project onto a *random* linear subspace of the same rank as the cross-covariance matrix. To reduce the variance of our results, we sample a fresh subspace for each minibatch, and erase that subspace at each layer, reporting the cross-entropy loss averaged over subspaces.

**Training efficiency.** Algorithm 1 avoids redundant computation by caching the layer $i$ hidden states for *every* data point, then using them to run layer $i + 1$. This approach has the downside of requiring a large amount of memory or disk space during training (up to 500GB in our experiments). It's possible to avoid caching any hidden states and instead recompute them as needed, at the expense of increasing the total compute cost from $O(\ell)$ to $O(\ell^2)$.

## 6.2 Results

We find strong evidence that autoregressive language models heavily rely on linearly encoded part-of-speech information. While erasing a randomly selected subspace has little to no effect on language modeling performance, scrubbing away part-of-speech information induces a large increase in perplexity across all models (Table 1).

The specific numbers, however, depend on the erasure method used: SAL induces significantly larger increases in perplexity for all models we tested. We take this to mean that SAL inflicts more collateral damage on other useful features in the representation than LEACE does. In other words, interventions made with LEACE are more *surgical* than those made with prior work; they more closely approximate the ideal of a perfect intervention which only erases the target concept and keeps everything else fixed [40, 15]. If this experiment were conducted with SAL alone, we would have *overestimated* the causal effect of part-of-speech.

## 7 Limitations and Future Work

Much work remains to be done to validate concept scrubbing. Specifically, we'd like to see experiments that target concepts much narrower than part-of-speech, and use behavioral metrics to determine whether scrubbing changes the network in the ways we'd intuitively expect. If these experiments succeed, an exciting next step would be the incorporation of concept scrubbing into the pretraining and/or finetuning process. This may make it possible to train deep neural networks subject to *conceptual constraints*. It remains to be seen if gradient-based optimizers will be able to "circumvent" such constraints by learning completely nonlinear representations of protected attributes.

In this work, we focused exclusively on *linear* concept erasure due to its simplicity and tractability. Some authors have proposed nonlinear concept erasure techniques based on kernel methods, but have found that erasure functions fit using one kernel do not generalize well to other kernels [33, 37]. We conjecture that it is intractable to nondestructively edit X so as to prevent a general nonlinear adversary from recovering Z, unless the data generating process for X is known in detail.[6]

A major motivation of concept erasure is that it promises to prevent models from using a concept in a *post hoc*, model-agnostic fashion. But if our concept scrubbing procedure turns out to yield unsatisfactory results in practical use cases, the most promising research direction might then be to improve model-*specific* techniques, such as those that modify the training procedure [8, 9, 14].

## 8 Acknowledgements

We are grateful to CoreWeave for providing the compute resources used in Section 6. Shauli Ravfogel is grateful to be supported by the Bloomberg Data Science PhD Fellowship.

---

[6]We suspect erasing a concept is at least as hard as extracting it from the original representation. But in the worst case, information about Z could be encoded *cryptographically* in X, which would be intractable to decode given standard computational complexity assumptions. If the data is generated by a known algorithm, however, it may be possible to efficiently eliminate mutual information between Z and X by simply breaking the links in the causal graph that connect them.

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

# A   Additional Related Work

The problem of linear concept erasure is an instance of the general problem of information removal. Information removal methods generally divide into adversarial methods, which are applied during training, and the post-hoc linear methods considered in this paper. Adversarial methods [8, 42, 4, 9, 44] use a gradient-reversal layer during training to induce representations that do not encode the protected attribute. However, Elazar and Goldberg [9] have shown that these methods fail in exhaustively removing all the information associated with the protected attribute: it is often possible to train new adversaries that successfully recover the removed information. Linear methods have been proposed as a tractable alternative, where one identifies a linear subspace that captures the concept of interest, and neutralizes it using algebraic techniques. Different methods have been proposed for the identification of the subspace, e.g. PCA and variants thereof [3, 20], orthogonal-rotation [7], classification-based [29], spectral [37, 36] and adversarial approaches [32].

Few works theoretically characterize the condition of linear guardedness. Haghighatkhah et al. [16] extensively analyzed the problem of preventing linear classification, with the focus on decreasing accuracy. They provide a constructive proof of an optimal intervention for an SVM classifier. Ravfogel et al. [31] have proposed a formal definition of linear guardedness based on $\mathcal{V}$ information, and characterized the fairness implications of guardedness; we show the relations with our definition above. Ravfogel et al. [32] provide an adversarial formulation of the problem, derive a closed-formed solution to certain cases, and propose an SGD-based optimization for others. While they seek an *orthogonal* projection, we empirically showed that their solution is very close to ours. Sadeghi et al. [35] and Sadeghi and Boddeti [34] both study an adversarial formulation of concept erasure for linear regression, and they trade-off with main-task performance. In contrast to Ravfogel et al. [32], they consider a general linear adversary, i.e. not necessarily a projection matrix. Closest to our work are Kleindessner et al. [20], Haghighatkhah et al. [17], Shao et al. [37]. As we showed above (§ 4), those methods do achieve the goal of linear guardedness though they are unable to prove this fact. At the same time, they are not optimal in terms of damage to the original representation space.

# B   Equivalence of Guardedness with the Optimality of Constant Predictors

The following two theorems establish the equivalence of conditions 1 and 2 (indeed, they do so in the general setting, with no assumption of convex loss or linear predictors).

**Theorem B.1.** *Suppose* X *$(\mathcal{V}, \mathfrak{L})$-guards* Z. *Then for every loss $\mathcal{L} \in \mathfrak{L}$, the corresponding trivially attainable loss $L_\tau^{(Z,\mathcal{L})}$ cannot be improved upon by any predictor $\eta(\cdot; \boldsymbol{\theta}) \in \mathcal{V}$, i.e. $L_\tau = \inf_{\boldsymbol{\theta}} \mathbb{E}[\mathcal{L}(\eta(X; \boldsymbol{\theta}), Z)]$.*

*Proof.* Consider the null random vector $X'(\omega) = \mathbf{0}$. Since all predictors are constant on $X'$, and the trivially attainable loss gives the *best* available expected loss among constant predictors, we must have:

$$L_\tau = \inf_{\boldsymbol{\theta}} \mathbb{E}[\mathcal{L}(\eta(X'; \boldsymbol{\theta}), Z)] \tag{4}$$

The right side of equation (4) is the best possible loss achievable by a function $\eta(\cdot; \boldsymbol{\theta})$ on the joint distribution of $(X', Z)$, which by the definition of guardedness is upper bounded by the best possible loss achievable on the joint distribution of $(X, Z)$:

$$\inf_{\boldsymbol{\theta}} \mathbb{E}[\mathcal{L}(\eta(X'; \boldsymbol{\theta}), Z)] \leq \inf_{\boldsymbol{\theta}} \mathbb{E}[\mathcal{L}(\eta(X; \boldsymbol{\theta}), Z)] \tag{5}$$

Combining equations (4) and (5), and the fact that all constant functions exist in our function class $\mathcal{V} = \{\eta(\cdot; \boldsymbol{\theta})\}$, we arrive at our desired result:

$$L_\tau = \inf_{\boldsymbol{\theta}} \mathbb{E}[\mathcal{L}(\eta(X; \boldsymbol{\theta}), Z)]$$

□

**Theorem B.2.** *Suppose that for every loss $\mathcal{L} \in \mathfrak{L}$, the corresponding trivially attainable loss $L_\tau^{(Z,\mathcal{L})}$ cannot be improved upon by any predictor $\eta(\cdot; \boldsymbol{\theta}) \in \mathcal{V}$, i.e. $L_\tau = \inf_{\boldsymbol{\theta}} \mathbb{E}[\mathcal{L}(\eta(X; \boldsymbol{\theta}), Z)]$. Then X $(\mathcal{V}, \mathfrak{L})$-guards Z.*

*Proof.* Let $X' : \Omega \to \mathbb{R}^d$ be any other random data vector with finite first moment.

Since all constant predictors exist in our predictor class $\mathcal{V} = \{\eta(\cdot; \boldsymbol{\theta})\}$, the best loss achievable on $(X', Z)$ by functions in $\mathcal{V}$ must be at least as good as the trivially attainable loss (the best loss available by such constant predictors):

$$\inf_{\boldsymbol{\theta}} \mathbb{E}[\mathcal{L}(\eta(X'; \boldsymbol{\theta}), Z)] \leq L_\tau$$

By assumption, the trivially attainable loss cannot be improved upon over $(X, Z)$ by predictors in $\mathcal{V}$:

$$L_\tau = \inf_{\boldsymbol{\theta}} \mathbb{E}[\mathcal{L}(\eta(X; \boldsymbol{\theta}), Z)]$$

Since our choice of $X'$ was arbitrary, this shows that X maximizes the minimal achievable loss, so X $(\mathcal{V}, \mathfrak{L})$-guards Z. $\qquad \square$

## C  Linear Guardedness is Equivalent to Linear Statistical Parity

To measure the effect of linear guardedness on main-task classifiers, we use the following minimal definition of "fairness" with respect to an attribute, adapted from Edwards and Storkey [8].

**Definition C.1** (Statistical Parity). *Let* X *and* Z *be defined as above, and let* $f$ *be a function with domain* $\mathbb{R}^d$. *Then* $f$ *exhibits **statistical parity** with respect to* Z *when evaluated on* X *if*

$$\forall z \in \mathcal{Z} : \mathbb{E}[f(X)|Z = z] = \mathbb{E}[f(X)].$$

We now prove the equivalence of conditions 3 and 5.

**Theorem C.2.** *Let* X *and* Z *be defined as above. Then every linear predictor* $f(\mathbf{x}) = \mathbf{b} + \mathbf{W}\mathbf{x}$ *exhibits statistical parity w.r.t.* Z *when evaluated on* X *if and only if each class-conditional mean* $\mathbb{E}[X|Z = z]$ *is equal to* $\mathbb{E}[X]$.

*Proof.* Suppose each class-conditional mean $\mathbb{E}[X|Z = z]$ is equal to $\mathbb{E}[X]$. Then by the linearity of expectation, we have for all $z \in \mathcal{Z}$:

$$\mathbb{E}[f(X)|Z = z] = \mathbb{E}[\mathbf{W}X + \mathbf{b}|Z = z] = \mathbf{W}\mathbb{E}[X|Z = z] + \mathbf{b} = \mathbf{W}\mathbb{E}[X] + \mathbf{b} = \mathbb{E}[f(X)].$$

This matches the definition of statistical parity provided in Definition C.1.

Conversely, suppose every linear predictor $f(\mathbf{x}) = \mathbf{b} + \mathbf{W}\mathbf{x}$ exhibits statistical parity w.r.t. Z when evaluated on X. Then this holds for the identity function $\mathrm{id}(\mathbf{x}) = \mathbf{x}$, and thus for all $z \in \mathcal{Z}$:

$$\mathbb{E}[X|Z = z] = \mathbb{E}[\mathrm{id}(X)|Z = z] = \mathbb{E}[\mathrm{id}(X)] = \mathbb{E}[X].$$

$\qquad \square$

## D  Implications for Prior Work

In this section we discuss the implications of Theorem 4.1, which characterizes the necessary and sufficient conditions for an affine erasure function to yield a perfectly linearly guarded dataset, for methods proposed in prior work.

Spectral Attribute RemovaL (SAL) [37] uses the top $n$ left singular vectors of $\mathbf{\Sigma}_{XZ}$ to construct an orthogonal projection matrix $\mathbf{Q}_{\mathrm{SAL}} = \mathbf{I} - \mathbf{U}_{:n}\mathbf{U}_{:n}^T$ which is then applied to X. Notably, while $n$ is presented as a free parameter in their method, all of their experiments involve binary classification problems where Z is a one-hot vector, and $n$ is set to a value no greater than 2. We'll call the version of SAL where $n = \mathrm{rank}(\mathbf{\Sigma}_{XZ})$, "full-rank SAL." Since these left singular vectors are an orthonormal basis for $\mathbf{\Sigma}_{XZ}$'s column space, Theorem 4.1 implies that full-rank SAL guarantees linear guardedness.

Mean Projection (MP) [17] orthogonally projects X onto the orthogonal complement of the span of the difference in class centroids $\mathbb{E}[X|Z = 1] - \mathbb{E}[X|Z = 0]$, where Z is assumed to be binary. Since the centroids are equal after the projection, this method guarantees linear guardedness by Theorem 3.1. In fact, by Theorem 3.4, MP is mathematically equivalent to SAL when Z is a one-dimensional random vector taking one of two possible values.

# E Derivation of LEACE

**Theorem 4.2.** *Let* X *and* Z *be centered random vectors taking values in* $\mathbb{R}^d$ *and* $\mathbb{R}^k$ *respectively, each of finite second moment. Let* $\mathbf{M} \in \mathbb{R}^{d \times d}$ *be a p.s.d. matrix defining a (possibly degenerate) inner product on* $\mathbb{R}^d$: $\langle \mathbf{x}, \mathbf{y} \rangle_{\mathbf{M}} = \mathbf{x}^T \mathbf{M} \mathbf{y}$. *Let* $\boldsymbol{\Sigma}_{\mathrm{XX}} \in \mathbb{R}^{d \times d}$ *be* X*'s covariance matrix, and* $\boldsymbol{\Sigma}_{\mathrm{XZ}} \in \mathbb{R}^{d \times k}$ *be the cross-covariance matrix of* X *and* Z. *Let* $\mathbf{A}^+$ *denote the Moore-Penrose pseudoinverse of a matrix* $\mathbf{A}$, *and let* $\mathbf{A}^{1/2}$ *be the p.s.d. square root of a p.s.d. matrix* $\mathbf{A}$. *Then the objective*

$$\operatorname*{argmin}_{\mathbf{P} \in \mathbb{R}^{d \times d}} \mathbb{E}\left[\left\|\mathbf{P}\mathrm{X} - \mathrm{X}\right\|_{\mathbf{M}}^2\right] \quad \text{subject to } \mathrm{Cov}(\mathbf{P}\mathrm{X}, \mathrm{Z}) = \mathbf{0}$$

*has the following solution:*

$$\mathbf{P}^* = \mathbf{I} - \mathbf{W}^+ \mathbf{P}_{\mathbf{W}\boldsymbol{\Sigma}_{\mathrm{XZ}}} \mathbf{W},$$

*where* $\mathbf{W}$ *is the whitening transformation* $(\boldsymbol{\Sigma}_{\mathrm{XX}}^{1/2})^+$ *and* $\mathbf{P}_{\mathbf{W}\boldsymbol{\Sigma}_{\mathrm{XZ}}} = (\mathbf{W}\boldsymbol{\Sigma}_{\mathrm{XZ}})(\mathbf{W}\boldsymbol{\Sigma}_{\mathrm{XZ}})^+$ *is the orthogonal projection matrix onto* $\mathrm{colsp}(\mathbf{W}\boldsymbol{\Sigma}_{\mathrm{XZ}})$.

Below are two independent proofs of Theorem 4.2.

## E.1 Algebraic Proof

*Proof.* We shall first show that, in any orthonormal basis,[7] each row $\mathbf{P_i}$ constitutes an independent optimization problem, and then select a basis in which we can easily show that the corresponding component $\mathrm{X}_i$ of X can be almost surely decomposed into a linear combination of mutually uncorrelated components in the whitened random vector $\mathbf{W}\mathrm{X}$, some of which correlate with Z and some of which do not. The solution $(\mathbf{P}\mathrm{X})_i$ is then that same linear combination, restricted to those components which do not correlate with Z.

Consider first an orthonormal basis diagonalizing the inner product $\mathbf{M}$, so that $\langle \mathbf{x}, \mathbf{y} \rangle_{\mathbf{M}} = \sum_{i=1}^d \alpha_i x_i y_i$ for fixed $\alpha_1, \ldots, \alpha_d \geq 0$. This allows us to treat each row $\mathbf{P_i} \in \mathbb{R}^d$ of $\mathbf{P}$ as a separate optimization problem,

$$\operatorname*{argmin}_{\mathbf{P_i} \in \mathbb{R}^d} \mathbb{E}\left[\alpha_i \left(\mathbf{P_i}^T \mathrm{X} - \mathrm{X}_i\right)^2\right] \quad \text{subject to } \mathrm{Cov}(\mathbf{P_i}^T \mathrm{X}, \mathrm{Z}) = \mathbf{0},$$

at which point the weights $\alpha_i$ of each subproblem become irrelevant, and our objective may as well be Euclidean, allowing us to view each row as an independent optimization problem not just in this basis, but from any convenient one.

So now let $\ell = \mathrm{rank}(\boldsymbol{\Sigma}_{\mathrm{XZ}}) = \mathrm{rank}(\boldsymbol{\Sigma}_{\mathbf{W}\mathrm{X},\mathrm{Z}})$ and $m = \mathrm{rank}(\boldsymbol{\Sigma}_{\mathrm{XX}}) = \mathrm{rank}(\boldsymbol{\Sigma}_{\mathbf{W}\mathrm{X},\mathbf{W}\mathrm{X}})$, and consider a (new) orthonormal basis whose first $m$ coordinates span the column (and row) space of $\mathbf{W}$ (i.e. the subspace of $\mathbb{R}^d$ in which X and $\mathbf{W}\mathrm{X}$ have nonzero variance), and whose first $\ell \leq m$ coordinates span the column space of $\boldsymbol{\Sigma}_{\mathbf{W}\mathrm{X},\mathrm{Z}}$ (i.e. the subspace of $\mathbb{R}^d$ in which $\mathbf{W}\mathrm{X}$ has nonzero covariance with Z).

Any component of X can be (almost surely) written as a fixed linear combination of the nontrivial components of its whitening $\mathbf{W}\mathrm{X}$:

$$\mathrm{X}_i = (\mathbf{W}^+ \mathbf{W}\mathrm{X})_i = \sum_{j=1}^m W_{ij}^+ (\mathbf{W}\mathrm{X})_j. \qquad \text{(almost surely)}$$

Meanwhile, any component of $\mathbf{P}\mathrm{X}$ can be (always) written as a fixed linear combination of the nontrivial components of $\mathbf{W}\mathrm{X}$ and the almost surely zero components of X:

$$(\mathbf{P}\mathrm{X})_i = \sum_{j=1}^m A_{ij}(\mathbf{W}\mathrm{X})_j + \sum_{j=m+1}^d B_{ij}\mathrm{X}_j,$$

i.e. $\mathbf{P} = \mathbf{A}\mathbf{W} + \mathbf{B}\mathbf{V}$, where $\mathbf{V} = \mathbf{I} - \mathbf{W}^+\mathbf{W}$ is the orthogonal projection onto X's almost surely zero components.

---

[7] Throughout this proof, we abuse the notations $\mathrm{X}_i, \mathbf{P_i}$, etc. to refer to the $i^{\text{th}}$ component in the specified basis, not necessarily the standard one.

The $i^{\text{th}}$ sub-objective is then:

$$\mathbb{E}\big(\mathbf{P_i}^T\mathrm{X} - \mathrm{X}_i\big)^2 = \mathbb{E}\left[\sum_{j=1}^m (A_{ij} - W_{ij}^+)(\mathbf{W}\mathrm{X})_j\right]^2 = \sum_{j=1}^m (A_{ij} - W_{ij}^+)^2,$$

where we have safely ignored the almost surely zero terms $B_{ij}\mathrm{X}_j$ ($j > m$), and used the fact that the first $m$ components of $\mathbf{W}\mathrm{X}$ have identity covariance matrix.

$\mathbf{P}\mathrm{X}$ is almost surely equal to $\mathbf{AW}\mathrm{X}$, so our constraint $\mathrm{Cov}(\mathbf{P}\mathrm{X}, \mathrm{Z}) = \mathbf{0}$ is equivalent to $\mathbf{A}\boldsymbol{\Sigma}_{\mathbf{W}\mathrm{X},\mathrm{Z}} = \mathrm{Cov}(\mathbf{AW}\mathrm{X}, \mathrm{Z}) = \mathbf{0}$, i.e. $A_{ij} = 0$ when $j \leq \ell$, since the first $\ell$ components are those for which $\mathbf{W}\mathrm{X}$ correlates with $\mathrm{Z}$. Subject to this, the objective is minimized for $A_{ij} = W_{ij}^+$ when $j > \ell$, i.e. $\mathbf{A} = \mathbf{W}^+(\mathbf{I} - \mathbf{P}_{\mathbf{W}\boldsymbol{\Sigma}_{\mathrm{XZ}}})$.

The particular choice $\mathbf{B} = \mathbf{I}$ gives our solution $\mathbf{P}^* = \mathbf{I} - \mathbf{W}^+\mathbf{P}_{\mathbf{W}\boldsymbol{\Sigma}_{\mathrm{XZ}}}\mathbf{W}$, leaving the non-varying components of X intact (see Fig. 1 for a visualization). $\qquad\square$

The solution is unique except for columns corresponding to the components of X with zero variance, and rows corresponding to the zero-weighted components of the (pseudo) inner product $\mathbf{M}$.

### E.2 Covector Proof

*Proof.* We assume without loss of generality that vectors in $\mathbb{R}^d$ are represented in a basis diagonalizing the inner product $\mathbf{M}$, so that $\langle \mathbf{x}, \mathbf{y}\rangle_{\mathbf{M}} = \sum_{i=1}^d m_i x_i y_i$ for fixed $m_1, \ldots, m_d \geq 0$. This allows us to treat each row $\mathbf{P_i} \in \mathbb{R}^d$ of $\mathbf{P}$ as a separate optimization problem,

$$\operatorname*{argmin}_{\mathbf{P_i} \in \mathbb{R}^d} \mathbb{E}\Big[m_i\big(\mathbf{P_i}^T\mathrm{X} - \mathrm{X}_i\big)^2\Big] \quad \text{subject to } \mathrm{Cov}(\mathbf{P_i}^T\mathrm{X}, \mathrm{Z}) = \mathbf{0}.$$

Our objective only depends on $\mathbf{P_i}$ through its effect on the scalar random variable $\xi = \mathbf{P_i}^T\mathrm{X}$. All random variables[8] of the form $\zeta = \mathbf{u}_\zeta^T\mathrm{X}$ for some covector $\mathbf{u}_\zeta^T \in \mathbb{R}^d$ form a vector space $U$, which we equip with the covariance inner product $\langle \xi, \zeta\rangle_{\mathrm{Cov}} = \mathrm{Cov}(\xi, \zeta) = \mathbb{E}[\xi\zeta] = \mathbf{u}_\xi^T\boldsymbol{\Sigma}_{\mathrm{XX}}\mathbf{u}_\zeta$.

By the linearity of covariance, the elements of $U$ uncorrelated with Z form a subspace $Z^\perp \subseteq U$. Note also that $\xi \in Z^\perp$ if and only if $\xi$'s covector $\mathbf{u}_\xi^T$ satisfies $\mathrm{Cov}(\mathbf{u}_\xi^T\mathrm{X}, \mathrm{Z}) = \mathbf{u}_\xi^T\boldsymbol{\Sigma}_{\mathrm{XZ}} = \mathbf{0}_k$, and that these covectors themselves form the subspace $\mathrm{colsp}(\boldsymbol{\Sigma}_{\mathrm{XZ}})^\perp$ of $\mathbb{R}^d$.

Our objective now reduces to finding a covector $\mathbf{P}_i^T$ that defines the orthogonal projection of $\mathrm{X}_i$ onto $Z^\perp$. The difficulty is that orthogonality of elements in $U$ is not equivalent to orthogonality of the corresponding covectors. We can fix this by changing the basis in which covectors are represented. Since $\mathrm{X} \in \mathrm{colsp}(\mathbf{W})$ a.s., we can write any element of $U$ as a linear form in $\mathbf{W}\mathrm{X}$ rather than X by applying the change-of-basis $\mathbf{u}_\xi' = \mathbf{W}^+\mathbf{u}_\xi$ to every covector: $\xi = (\mathbf{u}_\xi')^T\mathbf{W}\mathrm{X} = \mathbf{u}_\xi^T\mathbf{W}^+\mathbf{W}\mathrm{X}$ a.s.

In this new basis, which is orthonormal under our covariance inner product, each component of X is written $\mathrm{X}_i = (\mathbf{W}^+)_i^T\mathbf{W}\mathrm{X}$ and the inner product of any two elements of $U$ is simply the Euclidean inner product of the corresponding covectors:[9]

$$\langle \xi, \zeta\rangle_{\mathrm{Cov}} = \mathrm{Cov}(\mathbf{u}_\xi'^T\mathbf{W}\mathrm{X}, \mathbf{u}_\zeta'^T\mathbf{W}\mathrm{X}) = \mathbf{u}_\xi'^T\mathbf{W}\boldsymbol{\Sigma}_{\mathrm{XX}}\mathbf{W}\mathbf{u}_\zeta' = \mathbf{u}_\xi'^T\mathbf{u}_\zeta'.$$

Since the two inner products are now equivalent, and $Z^\perp$ is precisely those random variables with covector $\mathbf{u}' \in \mathrm{colsp}(\mathbf{W}\boldsymbol{\Sigma}_{\mathrm{XZ}})^\perp$, the orthogonal projection of $\mathrm{X}_i$ onto $Z^\perp$ is also an orthogonal projection of its covector $(\mathbf{W}^+)_i^T$ onto $\mathrm{colsp}(\mathbf{W}\boldsymbol{\Sigma}_{\mathrm{XZ}})^\perp$:

$$\hat{\mathrm{X}}_i = (\mathbf{W}^+)_i^T(\mathbf{I} - \mathbf{P}_{\mathbf{W}\boldsymbol{\Sigma}_{\mathrm{XZ}}})(\mathbf{W}\mathrm{X}) \tag{6}$$

Putting all the components of X together, we have our final solution,

$$\hat{\mathrm{X}} = (\mathbf{I} - \mathbf{W}^+\mathbf{P}_{\mathbf{W}\boldsymbol{\Sigma}_{\mathrm{XZ}}}\mathbf{W})\mathrm{X},$$

which is almost surely equivalent to Eq. 6, but keeps the non-varying components of X intact. $\qquad\square$

---

[8]Strictly speaking, equivalence classes of almost surely equal random variables.

[9]If $\boldsymbol{\Sigma}_{\mathrm{XX}}$ is full rank, there is a one-to-one correspondence between random variables in $U$ and covectors. In the singular case, we may choose the component of the covector inside $\ker(\boldsymbol{\Sigma}_{\mathrm{XX}})$ arbitrarily, since it will make no difference to the inner product.

## F  The Optimality of Oblique Projections

As noted in subsection 4.2, the optimal affine erasure function $r(\mathbf{x}) = \mathbf{b} + \mathbf{P}\mathbf{x}$ does *not* in general use an orthogonal projection for the matrix $\mathbf{P}$. A simple example illustrates why. Let $d = 2, k = 1$ so that X takes values in $\mathbb{R}^2$ and Z takes values in $\mathbb{R}$, with the first feature $X_1$ and the label Z each independently and uniformly distributed in $\{-1, +1\}$, and the second feature $X_2$ simply equal to the sum $X_2 = X_1 + Z$. A dataset reflecting such a distribution has four $(\mathbf{x}, \mathbf{y})$ pairs:

$$([1, 2]^T, 1), \quad ([1, 0]^T, -1), \quad ([-1, 0]^T, 1), \quad ([-1, -2]^T, -1)$$

In this case, *all* of the information X has about Z resides in $X_2$, so the minimally disruptive orthogonal projection which guards Z will nullify that component:

$$\mathbf{P}_{\text{ortho}} = \begin{bmatrix} 1 & 0 \\ 0 & 0 \end{bmatrix}$$

On the other hand, $X_1$ contains some information about $X_2$ (despite having no information about Z), allowing a partial reconstruction of $X_2$ while preserving full concept erasure:

$$\mathbf{P}_{\text{oblique}} = \begin{bmatrix} 1 & 0 \\ 1 & 0 \end{bmatrix}$$

Both methods fully erase the ability to predict Z from the data, however a simple calculation shows the second, oblique method to perform better as measured by mean squared edit distance:

$$\mathbb{E}\|\mathbf{P}_{\text{ortho}}X - X\|^2 = 2, \quad \mathbb{E}\|\mathbf{P}_{\text{oblique}}X - X\|^2 = 1$$

## G  Equivalence of Guardedness Definitions

Xu et al. [43] define the **conditional $\mathcal{V}$-entropy** of Z given X as the lowest achievable cross-entropy loss predicting Z with a function of X in the predictor class $\mathcal{V}$. In our notation:

$$H_{\mathcal{V}}(Z \mid X) = \inf_{\theta \in \Theta} \mathbb{E}[\mathcal{L}(\eta(X; \boldsymbol{\theta}), Z)],$$

where $\mathcal{L}(\eta, z) = -\log \frac{\exp(\eta_z)}{\sum_{i=1}^{k} \exp(\eta_i)}$ is the cross-entropy loss function.

They then define the (unconditional) $\mathcal{V}$-**entropy** $H_{\mathcal{V}}(Z) = H_{\mathcal{V}}(Z \mid \mathbf{0})$ to be the lowest achievable cross-entropy loss in the case of a constantly null random data variable. This is exactly our trivially attainable loss $L_\tau$ (Definition 2.2).

Finally, they define the $\mathcal{V}$-**information** from X to Z as the reduction in $\mathcal{V}$-entropy as compared to using such a null random data variable:

$$I_{\mathcal{V}}(X \to Z) = H_{\mathcal{V}}(Z) - H_{\mathcal{V}}(Z \mid X).$$

Using these notions, Ravfogel et al. [31] say that X is $\epsilon$-**guarded** with respect to $\mathcal{V}$ if $I_{\mathcal{V}}(X \to Z) < \epsilon$.

In Appendix B, we showed the equivalence of guardedness (as we have defined it in Definition 2.1) to the optimality of the trivially attainable loss. That is, X $(\mathcal{V}, \mathfrak{L})$-guards Z when $H_{\mathcal{V}}(Z \mid X) = L_\tau = H_{\mathcal{V}}(Z)$, in the case where $\mathfrak{L}$ is the singleton class consisting solely of the cross-entropy loss function. In the language of [31], X is $\epsilon$-guarded with respect to $\mathcal{V}$ for all $\epsilon > 0$.

## H  Constraining Norm Growth

In early concept scrubbing experiments (Sec. 6), we found that at specific layers in some models, concept scrubbing with LEACE would cause the norm of the representation to diverge, leading to NaN outputs. By contrast, SAL never caused divergence, even though it causes a larger disruption to model performance on average (Table 1). This is because SAL uses an orthogonal projection $\mathbf{Q}$, whose eigenvalues are thus all in $\{0, 1\}$, so the norm of the hidden state can never increase after erasure, while LEACE's oblique projection matrix $\mathbf{P}$ does generally have singular values greater than 1. To combine the superior average-case MSE of LEACE with the stability of SAL, we adopt a simple regularization heuristic. After constructing $\mathbf{P}$, we analytically compute the

trace of the covariance matrix of the hidden states after applying $\mathbf{P}$. If $\mathrm{tr}(\mathbf{P}\boldsymbol{\Sigma}_{\mathrm{XX}}\mathbf{P^T}) > \mathrm{tr}(\boldsymbol{\Sigma}_{\mathrm{XX}})$, we solve a quadratic equation to find the convex combination $\mathbf{P'} = \alpha\mathbf{P} + (1 - \alpha)\mathbf{Q}$ such that $\mathrm{tr}(\boldsymbol{\Sigma}_{\mathrm{XX}}) = \mathrm{tr}(\mathbf{P'}\boldsymbol{\Sigma}_{\mathrm{XX}}(\mathbf{P'})^{\mathbf{T}})$. By Theorem 4.1, the set of matrices which ensure linear guardedness is convex,[10] so $\mathbf{P'}$ is guaranteed to be in the feasible set. Furthermore, since our mean squared error objective is convex, $\mathbf{P'}$ is guaranteed to have no worse MSE than $\mathbf{Q}$. We find this solves the divergence issue in practice.

# I    Oracle LEACE

The concept erasure method derived in Section 4 does not require access to concept labels at inference time. That is, we can fit an erasure function on a labeled training dataset, then apply the function to unlabeled datapoints. If we have oracle access to the label $z$ for each $x$, we can achieve an even more surgical edit. In Theorem I.1 below, we derive **Oracle LEACE**, a closed-form formula for the the nearest $X'$ to any $X$ such that $\mathrm{Cov}(X', Z) = \mathbf{0}$.

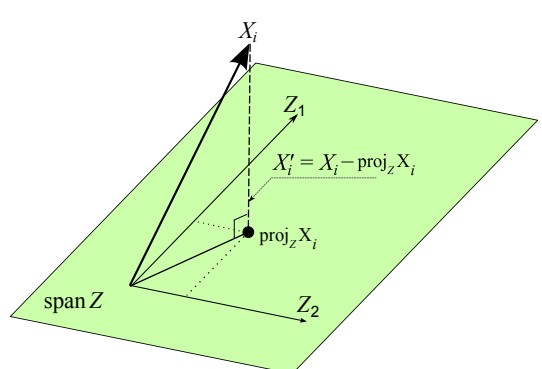

Figure 5: Orthogonal projection of $i^{\text{th}}$ component of X, itself a vector in the random variable Hilbert space $\mathcal{H}$, onto the span of the components of Z. The residual $X_i - \mathrm{proj}_{\mathcal{Z}}X_i$ is the closest vector to $X_i$ orthogonal to, and hence uncorrelated with, $\mathcal{Z} = \mathrm{span}(\{Z_1, Z_2\})$.

Like in Sec. 4, the resulting $X'_{\mathrm{LEACE}}$ is "nearest" to X with respect to all p.s.d. inner products $\mathbf{a}^T\mathbf{M}\mathbf{b}$ defined on $\mathbb{R}^d$ simultaneously. This is because, by expressing X in a basis that diagonalizes $\mathbf{M}$, we can decompose the problem into $d$ independent subproblems, one for each component of $X_i$. Each subproblem can then be viewed as an orthogonal projection, not in $\mathbb{R}^d$, but in an abstract vector space of real-valued random variables. For geometric intuition, see Figure 5.

Prior work has noted that computing an orthogonal projection in a random variable Hilbert space is equivalent to solving an ordinary least squares regression problem [1]. Our theorem is a natural extension of this work: we find that $X'_{\mathrm{LEACE}}$ is equal to the OLS residual from regressing X on Z, plus a constant shift needed to ensure that erasing Z does not change the mean of X.

**Theorem I.1** (Oracle Concept Erasure)**.** *Let $\mathcal{H}$ be the Hilbert space of square-integrable real-valued random variables equipped with the inner product $\langle \xi, \zeta \rangle_{\mathcal{H}} := \mathbb{E}[\xi\zeta]$. Let $(X, Z)$ be random vectors in $\mathcal{H}^d$ and $\mathcal{H}^k$ respectively. Then for every p.s.d. inner product $\langle \mathbf{a}, \mathbf{b} \rangle_{\mathbf{M}} = \mathbf{a}^T\mathbf{M}\mathbf{b}$ on $\mathbb{R}^d$, the objective*

$$\underset{X' \in \mathcal{H}^d}{\mathrm{argmin}}\ \mathbb{E}\big\|X' - X\big\|^2_{\mathbf{M}} \quad \text{subject to}\ \mathrm{Cov}(X', Z) = \mathbf{0}$$

*is minimized by the (appropriately shifted) ordinary least squares residuals from regressing X on Z:*

$$X'_{\mathrm{LEACE}} = X - \boldsymbol{\Sigma}_{\mathrm{XZ}}\boldsymbol{\Sigma}_{\mathrm{ZZ}}^+\big(Z - \mathbb{E}[Z]\big).$$

*Proof.* Assume w.l.o.g. that X and X' are represented in a basis diagonalizing $\mathbf{M}$, so we may write

$$\mathbb{E}\big\|X' - X\big\|^2_{\mathbf{M}} = \sum_{i=1}^{d} m_i\, \mathbb{E}\big[(X'_i - X_i)^2\big],$$

where $m_1, \ldots, m_d \geq 0$ are eigenvalues of $\mathbf{M}$. Crucially, each term in this sum is independent from the others, allowing us to decompose the primal problem into $d$ separate subproblems of the form $\|X'_i - X_i\|^2_{\mathcal{H}}$, one for each component $i$ of $(X, X')$.

**Factoring out constants.**    Now consider the subspace $\mathcal{C} = \mathrm{span}(1) \subset \mathcal{H}$ consisting of all constant (i.e. zero variance) random variables. Orthogonally decomposing $X_i$ along $\mathcal{C}$ yields $X_i = \tilde{X}_i + \mu_i$, where $\mu_i = \mathbb{E}[X_i] \in \mathcal{C}$ and $\tilde{X}_i = X - \mathbb{E}[X]_i \in \mathcal{C}^\perp$, and likewise for $X'_i$. Our objective is now

$$\big\|X'_i - X_i\big\|^2_{\mathcal{H}} = \big\|\mu'_i - \mu_i\big\|^2_{\mathcal{H}} + \big\|\tilde{X}'_i - \tilde{X}_i\big\|^2_{\mathcal{H}}. \tag{7}$$

---

[10]In fact, it is a subspace of $\mathbb{R}^{d \times d}$. For any matrices $\mathbf{A}, \mathbf{B} \in \mathbb{R}^{d \times d}$ such that $\mathbf{A}\boldsymbol{\Sigma}_{\mathrm{XZ}} = \mathbf{0}$ and $\mathbf{B}\boldsymbol{\Sigma}_{\mathrm{XZ}} = \mathbf{0}$, we have by linearity $(\alpha\mathbf{A} + \beta\mathbf{B})\boldsymbol{\Sigma}_{\mathrm{XZ}} = \alpha\mathbf{A}\boldsymbol{\Sigma}_{\mathrm{XZ}} + \beta\mathbf{B}\boldsymbol{\Sigma}_{\mathrm{XZ}} = \alpha\mathbf{0} + \beta\mathbf{0} = \mathbf{0}$ for any scalars $\alpha$ and $\beta$.

Since $\mu_i'$ and $\mu_i$ are orthogonal to $\tilde{X}_i'$ and $\tilde{X}_i$, and the constraint $\text{Cov}(X', Z) = \mathbf{0}$ is invariant to constant shifts, we can optimize the two terms in Eq. 7 independently. The first term is trivial: it is minimized when $\mu_i' = \mu_i$, and hence $X_i' = \tilde{X}_i + \mathbb{E}[X_i]$.

**Orthogonal projection.** We can now rewrite the zero covariance condition as an orthogonality constraint on $\tilde{X}_i$. Specifically, for every $i \in \{1 \dots d\}$ we have

$$\underset{\tilde{X}_i' \in \mathcal{H}}{\arg\min} \, \big\| \tilde{X}_i' - \tilde{X}_i \big\|_{\mathcal{H}}^2 \quad \text{s.t.} \ \forall j \in \{1 \dots k\} : \langle \tilde{X}_i', \tilde{Z}_j \rangle_{\mathcal{H}} = 0, \tag{8}$$

where $\tilde{Z} = Z - \mathbb{E}[Z]$. In other words, we seek the nearest $\tilde{X}_i'$ to $\tilde{X}_i$ orthogonal to $\mathcal{Z} = \text{span}(\{\tilde{Z}_1, \dots, \tilde{Z}_k\})$, which is simply the orthogonal projection of $\tilde{X}_i$ onto $\mathcal{Z}^\perp$. This in turn is equal to the ordinary least squares residual from regressing $\tilde{X}$ on $\tilde{Z}$:

$$\tilde{X}_i' = \tilde{X}_i - \text{proj}\big(\tilde{X}_i, \mathcal{Z}\big) = X_i - (\mathbf{\Sigma}_{XZ})_i \mathbf{\Sigma}_{ZZ}^+ (Z - \mathbb{E}[Z]) - \mathbb{E}[X_i]. \tag{9}$$

**Putting it all together.** Plugging Eq. 9 into $X_i' = \tilde{X}_i' + \mathbb{E}[X_i]$ and combining all components into vector form yields

$$X_{\text{LEACE}}' = X - \mathbf{\Sigma}_{XZ} \mathbf{\Sigma}_{ZZ}^+ (Z - \mathbb{E}[Z]), \tag{10}$$

which completes the proof. $\qquad\square$

## J  Notation Key

| | |
|---|---|
| $\mathcal{Z}$ | The space of one-hot labels $\{(z_1, \dots z_k) \in \{0, 1\}^k \mid \sum_{j=1}^k z_j = 1\}\}$ (treated interchangeably with the integers $\{1, \dots, k\}$ when convenient). |
| $X, Z$ | Integrable (i.e. finite first moment) random vectors taking values in $\mathbb{R}^d$ and $\mathbb{R}^k$ respectively (or their realized values inside an expectation, e.g. in $\mathbb{E}[f(X)]$). $Z$ is sometimes restricted to the one-hot labels $\mathcal{Z}$, in which case we assume each $\mathbb{P}(Z = j) > 0$. |
| $X_i, Z_j$ | The $i^{\text{th}}$ and $j^{\text{th}}$ components thereof, themselves scalar random variables (or their realized values inside an expectation). |
| $\xi, \zeta$ | Scalar random variables taking values in $\mathbb{R}$. |
| $\eta$ | A predictor function $\mathbb{R}^d \to \mathcal{Z}$ (or its value $\eta(X)$ when inside an expectation). |
| $\mathcal{V}$ | A space of predictor functions $\{\eta(\cdot; \boldsymbol{\theta}) : \mathbb{R}^d \to \mathbb{R}^k \mid \boldsymbol{\theta} \in \Theta\}$, parameterized by $\boldsymbol{\theta}$ and containing all constant functions. |
| $\mathfrak{L}$ | A space of loss functions $\{\mathcal{L} : \mathbb{R}^k \times \mathcal{Z} \to [0, \infty)\}$. |
| $r$ | An erasure function $\mathbb{R}^d \to \mathbb{R}^d$, hopefully making a minimal edit to $X$ that eliminates the ability to predict labels $Z$ with predictors in $\mathcal{V}$. |
| $\mathbf{A}$ | A matrix with entries in $\mathbb{R}$. |
| $A_{ij}$ | The entry thereof at the $i^{\text{th}}$ row and $j^{\text{th}}$ column. |
| $\mathbf{A}^+$ | The Moore-Penrose pseudoinverse of $\mathbf{A}$. |
| $\mathbf{v}$ | A column vector with entries in $\mathbb{R}$. |
| $v_i$ | The $i^{\text{th}}$ component thereof. |

