# OpenReview forum: "LEACE: Perfect linear concept erasure in closed form"
_NeurIPS.cc/2023/Conference — NeurIPS 2023 poster_

### Official Review · Reviewer_uqXm · 2023-06-25

**Soundness:** 3 good
**Presentation:** 2 fair
**Contribution:** 3 good
**Rating:** 4
**Confidence:** 4

**Summary:**

The paper presents a concept erasure procedure that also takes into account the distortion of the representations during the debiasing process. To this end, the paper builds on several works on iterative nullspace projection approaches and incorporates a regularization component. First, LEACE improves upon previous projection approaches by showing that E[X|Z]=E[X] is sufficient to protect information against a linear adversary. Using this result,  the paper presents a closed-form solution for the objective function. The paper shows experiments on debiasing and interoperability setups. The results showcase that LEACE can achieve similar levels of debiasing performance with the learned representations having much less distortion (measured in terms of its rank).

**Strengths:**

Strengths:

1. The paper proves the interesting result that achieving the following condition: E[X|Z]=E[X] results in preventing any linear adversary from extracting information about the protected attribute from a representation set.
2. The paper presents a closed-form optimal solution to linear debiasing with the L2 regularization constraint.
3. The paper shows experiments in several debiasing and interpretability setups. The results show that LEACE is able to achieve debiasing performance at par with previous approaches while retaining significant information (in terms of rank).

**Weaknesses:**

Weakness:

1. Is it unclear from the paper why the MSE reconstruction loss E[X-\hat{X}] is a good measure of information being retained? Ideally, I believe that the goal of concept erasure is to retain within-class information for each of the protected class categories. For such a requirement, there can be a trivial baseline where E[X|Z]=E[X] is achieved by normalizing the data within each class to have the same mean. Does this baseline work at all in terms of debiasing? If yes, what are the advantages of LEACE over this baseline? If no, is E[X|Z]=E[X] the right condition to be achieved?
2. The derived closed-form result looks very similar to the result in RLACE. It would be interesting to have an analytical discussion about the differences and their implications on debiasing performance.
3. Although having minimal edits is one of the key focuses of this paper, reconstruction loss is not reported in the results section. Simply reporting the rank is not convincing enough to evaluate the efficacy of the proposed approach in the paper. How does the reconstruction loss compare with other linear debiasing methods? If rank is the right metric to look at instead of the MSE loss, the paper needs more justification about why that is the case.
4. The fair representation learning framework proposed in this paper has been studied by various works previously. For example, LAFTR[1] and [2] have a similar reconstruction loss along with a classifier and adversary. The paper should have these baselines by simply removing the classifier component and having just the reconstruction and a linear adversary loss. This would be useful in understanding the efficacy of projection-based approaches over adversarial ones.
5. The experimental section is fairly weak at the moment. For example, in Table 1, why is LEACE not compared with RLACE and INLP in this setup? The authors do not report the performance of word embedding debiasing, which is one of the most common benchmarks for evaluating concept erasure approaches.
6. The organization/writing of the paper can be improved in several areas. Broadly, some of the theoretical results can be condensed and more context about the debiasing setup would be helpful for readers not too familiar with concept erasure literature. See some minor additional comments in the suggestions/limitations section.

[1]  https://arxiv.org/pdf/1802.06309.pdf

[2]  https://www.cs.toronto.edu/~toni/Papers/icml-final.pdf

**Questions:**

See the weakness section.

**Limitations:**

Line 45: “taking values in Rd”: what do values refer to here? I get that it refers to X but should be clearly stated.

Line 53: could not find the 4th variable defined in Section 2.1. Is it referring to the loss function?

Line 55: it's unclear why the family of loss functions needs to be used here. The guarding is defined w.r.t. to a single loss function.

The framing of different lemmas can be smoothened out to make it more straightforward.

More context about the functioning of the baseline approaches would be helpful.

---

> ### Author Rebuttal · Authors · 2023-08-10
>
> > 1. Is it unclear from the paper why the MSE reconstruction loss E[X-\hat{X}] is a good measure of information being retained?
>
> We agree with the reviewer that mean squared Euclidean distance is a limited measure. Luckily, we have since proven that LEACE is highly robust to the specific distance function used. Specifically, we prove that LEACE simultaneously minimizes the entire family of squared (pseudo-)norms of the form $\\| x \\|_{\mathbf M}^2 = x^T \mathbf{Mx}$ for some p.s.d. matrix $\mathbf M$. Intuitively, this means that the “weight” or “importance” assigned to each direction in $\mathbb R^d$ does not matter. See our general rebuttal for a proof of this result.
>
> We agree that retention of within-class information is necessary for practically useful concept erasure, but it is not sufficient for our purposes. This is because our primary aim is to minimize damage to the network’s input-output behavior, without retraining or fine tuning its parameters. Arguably a good proxy for this is the Fisher information metric, which locally approximates the KL divergence between the network’s final output before and after perturbing its activations. Since this metric is p.s.d., LEACE minimizes it “out of the box,” without any change to its formula. Thus, LEACE can indeed be viewed as approximately minimizing damage to the network’s behavior, even though it is architecture-agnostic.
>
> > For such a requirement, there can be a trivial baseline where E[X|Z]=E[X] is achieved by normalizing the data within each class to have the same mean. Does this baseline work at all in terms of debiasing? If yes, what are the advantages of LEACE over this baseline?
>
> Theorem 3.1 does indeed imply that the normalization technique you suggest ensures linear guardedness. The primary benefit of LEACE over mean-normalization is that mean-normalization requires access to ground truth concept labels at test time. This is because one needs to know the label in order to know which class centroid $\mathbb E[\mathrm X | \mathrm Z = i]$ should be subtracted from a given data point $\boldsymbol x$. We often lack ground-truth labels at test time, such as when de-biasing a classifier w.r.t. a protected attribute. Attempting to “guess” the label from observed attributes is likely to be noisy, and may itself be an instance of bias or unfairness.
>
> > 2. The derived closed-form result looks very similar to the result in RLACE. It would be interesting to have an analytical discussion about the differences and their implications on debiasing performance.
>
> We presume the reviewer is referring to Proposition 3.1 in Ravfogel et al. (2022) Linear Adversarial Concept Erasure, which proves that the orthogonal projection matrix onto $\mathrm{colsp}(\Sigma_{XZ})^\perp$ can be used to achieve guardedness w.r.t. linear regression models. Indeed, Theorem 4.1 implies that this formula achieves guardedness w.r.t. linear classifiers as well. But unlike LEACE, it is not surgical, since it does not take into account anisotropy in the covariance matrix of X (see Figure 1 in the rebuttal PDF), nor does it account for the possibility that X is not centered at the origin.
>
> > 3. Although having minimal edits is one of the key focuses of this paper, reconstruction loss is not reported in the results section.
>
> We thank the reviewer for noting this omission. We have therefore included a plot in our rebuttal PDF (Figure 2). We plan to replace Figure 1 in the current submission with this new plot in the camera-ready version.
>
> > The fair representation learning framework proposed in this paper has been studied by various works previously. For example, LAFTR[1] and [2] have a similar reconstruction loss along with a classifier and adversary. The paper should have these baselines by simply removing the classifier component and having just the reconstruction and a linear adversary loss.
>
> We don’t view our work as being directly comparable to the cited works “LAFTR” (Madras et al. 2018) and Zemel et al. (2013), which are both comprehensive proposals for learning fair representations. By contrast, LEACE and concept scrubbing are modular tools which can be applied in a post hoc manner to pre-trained models of all kinds.
>
> > The experimental section is fairly weak at the moment. For example, in Table 1, why is LEACE not compared with RLACE and INLP in this setup?
>
> We exclusively compare to SAL in the concept scrubbing experiment because it is the only applicable prior art which achieves perfect linear guardedness (by Theorem 4.1), and because it is efficiently scalable to very large datasets, such as the Pile and RedPajama pre-training corpora. This scalability is due to the fact that, like LEACE, SAL only depends on covariance and cross-covariance statistics which can be computed in a streaming fashion. RLACE and INLP, by contrast, require expensive gradient-based optimization and would be much more computationally intensive and time-consuming to run for this large-scale experiment.
>
> > could not find the 4th variable defined in Section 2.1. Is it referring to the loss function?
>
> $\mathfrak L$ is a family of loss functions $\mathcal L$, and its definition was missing from Sec. 2.1. We will correct this in the camera-ready version.
>
> > it's unclear why the family of loss functions needs to be used here. The guarding is defined w.r.t. to a single loss function.
>
> In our work, we define guardedness w.r.t. a family of loss functions, since our Theorem 3.1 applies to the family of all convex losses.
>
> > lack of experiments on uncontextualized word embeddings.
>
> While the suggestion is well taken, we believe this is beyond the scope of the current paper. We believe the current content and focus fully occupies a complete paper, and that further expansions to problems outside of the deep classification context would make the paper unwieldy and would require comparing against two different lines of literature simultaneously.

---

### Official Review · Reviewer_3Jip · 2023-07-06

**Soundness:** 3 good
**Presentation:** 4 excellent
**Contribution:** 3 good
**Rating:** 7
**Confidence:** 5

**Summary:**

This paper introduces a novel method to guard pretrained representations of deep neural networks from linear recovery of sensitive attributes Z. The notion of linear guardedness comes from the previous literature, while this paper proposes a simple characterisation when it takes place, which allows to dramatically simplify the algorithm as well as improve the performance.

Unlike the previous methods such as INLP, the LEACE projection matrix has rank d - k, where d is the dimension of the representation at hand, and k is the number of values Z can take (i.e. 2 in the binary case). This allows to guard the sensitive attribute with minimal losses. The authors confirm this with two experiments: 1) guarding gender in bios from De-Arteaga et al. 2) amnesic probing from Elazar et al.

pros:
- simple and effective method that outperforms previous literature
- well written, and the equivalence theorems are easy to follow
- the experiment in Figure 3 is fascinating, in particular, the conclusion that the information is concentrated in layer 11

cons:
- lack of error bars. For example, in the experiment corresponding to Figure 1, what happens if we feed a limited data into the LEACE algorithm and test guardedness on the rest?
- in the experiments the data to which LEACE is applied is the same as it is tested on. It is not tested whether the concept can be erased using dataset that is collected independent of the downstream task.
- the conclusion in section 5.3 is relevant to interpretation methods such as Kim et al "Interpretability beyond attribution: TCAV", who are also concerned with what kinds of information a neural network extracts at each layer. On the contrary to your claim, they report that the amount of "concept information" does not reduce as data passes from layer to layer, see Figure 5. I wonder how it compares to you measurements in the case of POS concept. Their method is very simple and should be easy to implement.
- it is also not entirely clear how it affects the remaining information. For example, if I look at two identical bios which differ only in the gender of a person, how would their representations compare after you have removed the gender concept with LEACE?

minor comments:

- Figure 3: ramdom
- l306: which a we

**Strengths:**

-

**Weaknesses:**

-

**Questions:**

-

**Limitations:**

-

---

> ### Author Rebuttal · Authors · 2023-08-10
>
> > lack of error bars.
>
> In Figure 1, the vertical lines crossing through each data point are 95% confidence intervals. We apologize that we did not make this clearer in the original submission and we intend to replace the error lines with a translucent error ribbon in the camera-ready version.
>
> > For example, in the experiment corresponding to Figure 1, what happens if we feed a limited data into the LEACE algorithm and test guardedness on the rest? …in the experiments the data to which LEACE is applied is the same as it is tested on. It is not tested whether the concept can be erased using dataset that is collected independent of the downstream task.
>
> We agree that examining the generalization properties of LEACE is an important research direction. There are many different metrics that could be used to measure generalization performance, and many different types of distribution shift that could be tested. Given the richness of the topic, we decided that we could not do it justice given the time and space constraints of this paper. We are excited to see this issue addressed in future work.
>
> > the conclusion in section 5.3 is relevant to interpretation methods such as Kim et al "Interpretability beyond attribution: TCAV", who are also concerned with what kinds of information a neural network extracts at each layer.
>
> Thank you for bringing this work to our attention. We would like to emphasize that, unlike Kim et al. (2018), the experiments in Sections 5 and 6 are not concerned with measuring the kinds of information extracted by the neural network at each layer. We are rather interested in the causal contribution of a query concept— in this case, part of speech— to the network’s performance. This is a distinct, albeit related quantity.
>
> Since the publication of Kim et al. (2018), several papers have found that the ease of extracting a concept from neural network activations is a poor proxy for the causal contribution of that concept to network behavior; see e.g. “Designing and Interpreting Probes with Control Tasks” by Liang et al. (2019), or “Probing classifiers: Promises, shortcomings, and advances” by Belinkov (2022) for a thorough literature review. We will clarify this issue in the camera-ready with appropriate citations.
>
> > it is also not entirely clear how it affects the remaining information. For example, if I look at two identical bios which differ only in the gender of a person, how would their representations compare after you have removed the gender concept with LEACE?
>
> We encourage the reviewer to examine Figure 1 in the PDF attached to our general rebuttal, which visualizes the LEACE erasure method on a 2D toy dataset.

---

> > ### Comment · Reviewer_3Jip · 2023-08-15
> >
> > Thank you for your response. I will keep the score unchanged for now.

---

### Official Review · Reviewer_EBcB · 2023-07-11

**Soundness:** 4 excellent
**Presentation:** 4 excellent
**Contribution:** 3 good
**Rating:** 7
**Confidence:** 3

**Summary:**

Suppose we are given a distribution of data points $(x,z)$, where $x \in \mathbb{R}^d$ and we have one-hot class labels $z \in \mathbb{R}^k$. This paper shows how to construct an affine transformation $\phi(x) = Px + b$ of the data so that
* $\phi(x)$ has zero covariance with $z$
* $\\|\phi(x) - x\\|$ is minimized, subject to the above condition

The paper argues that this is the optimal affine transformation that ``erases'' the concept $z$ from the representation $x$. Namely, linear regression on $x$ with any convex loss function will be unable to reconstruct $z$ better than random guessing. The paper empirically shows that the proposed method outperforms previous methods (RLACE, INLP) for erasing concepts from a representation, in terms of computational efficiency/collateral damage of the concept erasure.

Applications are demonstrated in: 1) fairness, and 2) probing language models to measure how much a certain concept matters to their performance.


**Strengths:**

The paper has excellent exposition: the proofs and definitions are clear and simple to follow, and the experiments are generally well explained. The applications to language models are compelling, and the paper demonstrates an improvement over previous approaches. The result is of broad interest to the community.

**Weaknesses:**

I am not an expert in the field of concept erasure, so I do not know if this method was previously known. However, my feeling is that the paper proposes a "trivial" solution: orthogonal projection of the random vector $x$ to the subspace of random vectors uncorrelated with $z$. So I find it surprising that this method was not previously known (maybe in a different field and under a different name).
Of course, this is not actually a weakness of the paper (but rather of the reviewer).

**Questions:**

1. The authors state that "prior work has focused solely on preventing linear models from leveraging Z", whereas their work applies also to deep neural networks. How does this claim relate to the cited papers [22, 5, 3, 37], which also intervene on the representations of deep neural networks?

2. Can the authors clarify the paragraph at the top of page 7, entitled "Results"? Why is 77.3% profession-prediction accuracy reported, and then 78.1% accuracy reported? What is the difference between these settings?


Typos:
* In Definition 2.1, \mathfrak{L} is said to have been defined in Section 2.1, but it is not.
* In Section 2.2, the equation between lines 56 and 57 is an argmax over X' but X' does not appear in the expression
* Figure 3, "ramdom" -> "random"
* Appendix E.1, "a forteriori" --> "a fortiori"

**Limitations:**

Yes

---

> ### Author Rebuttal · Authors · 2023-08-10
>
> >  my feeling is that the paper proposes a "trivial" solution: orthogonal projection of the random vector x to the subspace of random vectors uncorrelated with z. So I find it surprising that this method was not previously known (maybe in a different field and under a different name).
>
> We believe the simplicity of LEACE is one of its major strengths. After a thorough literature review, we were unable to find any prior work proposing an equivalent method, which suggests our approach may only seem “trivial” in hindsight.
>
> The closest prior art is the Spectral Attribute RemovaL (SAL) method of Shao et al. (2022), who erase a concept $\mathrm{Z}$ from a representation $\mathrm{X}$ by orthogonally projecting $\mathrm{X}$ onto $\mathrm{colsp}(\Sigma_{\mathrm{XZ}})^\perp$. Like SAL, LEACE also neutralizes the subspace $\mathrm{colsp}(\Sigma_{\mathrm{XZ}})$ with a projection. But unlike SAL, the LEACE projection is oblique (i.e. non-orthogonal) in general.
>
> To see the difference, we encourage the reviewer to examine Figure 1 in the PDF attached to our general rebuttal. SAL would orthogonally project the blue and orange points onto the dashed line, which does indeed achieve linear guardedness. But this is not the least-squares optimal solution, since it doesn’t respect the covariance structure of the data. Prior to applying SAL, the data has more variance along the x-axis than along the y-axis, but SAL inverts this structure. By contrast, LEACE preserves the original data distribution as best it can.
>
> We note also that our theorems prove the sufficiency of these methods for linear guardedness w.r.t. all convex loss functions (Theorem 3.1). Previously this had only been established for specific loss functions, such as the L2 loss (Ravfogel et al. 2022, “Linear Adversarial Concept Erasure”).
>
> > The authors state that "prior work has focused solely on preventing linear models from leveraging Z", whereas their work applies also to deep neural networks. How does this claim relate to the cited papers [22, 5, 3, 37], which also intervene on the representations of deep neural networks?
>
> We apologize for the misleading wording— it would be better to say that prior work has focused primarily, but not exclusively, on linear models. We will correct this mistake in the camera-ready version. That said, 3 of the 4 cited papers (Chowdhury et al. 2022, Celikkanat et al. 2020, and Subramanian et al. 2021) apply concept erasure to frozen contextualized embeddings extracted from the final layer of a language model, and not to intermediate activations. We would classify these papers as concerning linear models, since they aim to prevent a linear classifier trained on frozen embeddings from using a concept. The other paper (Lasri et al. 2022) does apply INLP to selected intermediate layers in BERT, but does not attempt to erase a concept from multiple layers simultaneously, as our concept scrubbing method does. We hope this clarifies the novelty of our concept scrubbing method.
>
> > Can the authors clarify the paragraph at the top of page 7, entitled "Results"? Why is 77.3% profession-prediction accuracy reported, and then 78.1% accuracy reported? What is the difference between these settings?
>
> When a profession predictor is fit on the original unedited training set, it achieves 77.3% accuracy on a LEACE’d validation set— that is, a validation set on which we have applied LEACE to remove the concept of gender. When the predictor is fit on a LEACE’d training set, it achieves 78.1% accuracy on the LEACE’d validation set. The gap between these two figures corresponds to the mild distribution shift created by LEACE.
>
> We apologize that this was not clearer in our submission and will rephrase the paragraph for clarity in the camera-ready version.

---

> > ### Comment · Reviewer_EBcB · 2023-08-14
> > **Response**
> >
> > Thank you for the clarifications. I am happy to keep my score of Accept, and would like to congratulate the authors on the excellent work.

---

### Official Review · Reviewer_rafU · 2023-07-27

**Soundness:** 3 good
**Presentation:** 3 good
**Contribution:** 3 good
**Rating:** 6
**Confidence:** 3

**Summary:**

This paper studies concept removal from features. For linear classifiers, the authors prove several equivalent characterizations of linear guardedness (reducing the accuracy of any linear classifiers to the trivial accuracy). In particular, the features that achieve linear guardedness have zero covariance with the labels.

Equipped with the new characterizations on the linear guardedness, the authors propose a new formulation of concept removal. Specially, the formulation finds the linear map minimizing the Euclidean distance such that the features after the linear transformation are linearly guarded. The formulation yields a quadratic program with a single linear equality constraint, and thus enjoys a closed-form solution.

The author conducts three experiments to verify the effectiveness of the method. The experiments on removing gender information from BERT shows that the proposed method successfully removes gender bias while preserving the classification accuracy.
They also apply the proposed method removing part-of-speech tag information from language models to investigate the importance of POS tag on language modeling tasks.

**Strengths:**

The idea of minimizing the Euclidean distance seems natural. The characterization on the linear guardedness is sound and useful. Importantly, this paper reveals connections with previously published methods and thus may strengthen the understanding of concept erasure. For example, the theoretical characterization justifies that SAL, Mean Projection and Fair PCA all achieve linear guardedness. Moreover, it is interesting that RLACE has the same top eigenvector with the method proposed in this paper.


**Weaknesses:**

1. The writing overall is good but the structure of writing can be improved. Theorem 4.3 strictly includes Theorem 4.2 as a special case. There is little benefit to separately present Theorem 4.2. Thus, they should be merged. Additionally, I find most of the proofs are not that insightful and thus it's better to defer them to the appendix. The extra space can be used to expand the preliminaries section as I find the current presentation is not entirely clear for audience who's not familiar with the topic.
2. One experiment that I am curious about is the part-of-speech tag prediction accuracy after concept scrubbing. As LEACE only guarantees linear guardedness, it is unclear from Section 5 and 6 whether non-linear classifiers still preserves the concept after concept scrubbing.

Minor:
- Line 325: lineally -> linearly

**Questions:**

1. In Figure 1 left, why does random projection have even higher accuracy compared to no intervention? I would expect random projection to be the same or slightly worse than no intervention. Figure 1 right does show they have similar loss.
2. The definition 2.1 might have an issue. The maximization right after Line 56 should be over all possible joint distributions over features and labels that has marginal distribution the same as $Z$.

**Limitations:**

This authors have listed their limitations in the paper which have been addressed.

---

> ### Author Rebuttal · Authors · 2023-08-09
>
> We thank the reviewer for their feedback on the presentation of the paper, especially the proofs. We will revise the manuscript based on their feedback for the camera ready version.
>
> > One experiment that I am curious about is the part-of-speech tag prediction accuracy after concept scrubbing. As LEACE only guarantees linear guardedness, it is unclear from Section 5 and 6 whether non-linear classifiers still preserves the concept after concept scrubbing.
>
> If the class-conditional distributions $\mathcal{P}(\mathrm{X} | \mathrm{Z} = i) : i \in 1 \ldots k$ are all Gaussian with equal covariance matrices, then LEACE will indeed prevent non-linear classifiers from predicting $\mathrm{Z}$ using the scrubbed representation $\mathrm{X}'$. This is because the resulting distributions $\mathcal{P}(\mathrm{X}' | \mathrm{Z} = i) : i \in 1 \ldots k$ will all be equal, and therefore $\mathrm{X}'$ will not have any mutual information with $\mathrm{Z}$. In most cases, however, concept scrubbing will be unable to prevent non-linear classifiers from extracting information about $\mathrm{Z}$ at least to some extent.
>
> > In Figure 1 left, why does random projection have even higher accuracy compared to no intervention?
>
> We thank the reviewer for bringing this issue to our attention. We found a typographical error in our plotting code, which caused the “No Intervention” accuracy to be reported as 0.892 instead of the correct value of 0.982. The corrected plot is included in the PDF attached to the general rebuttal as Figure 3.
>
> > The definition 2.1 might have an issue. The maximization right after Line 56 should be over all possible joint distributions over features and labels that has marginal distribution the same as Z.
>
> Although the current text does state X and Z are “jointly defined,” we agree with the reviewer that this should be made more clear. We intend to fix this issue in the camera-ready version.

---

> > ### Comment · Reviewer_rafU · 2023-08-15
> >
> > Thank you for the response. I suggest the authors incorporate the reviews in the revision.

---

> > > ### Author Response · Authors · 2023-08-16
> > >
> > > Thank you for reading our rebuttal. We would like to draw your attention to the fact that no revisions are allowed until camea-ready stage (https://neurips.cc/Conferences/2023/PaperInformation/NeurIPS-FAQ). We absolutely will (and have already begun to) incorporate the reviews into our draft, and we are excited to share the improved camera ready version with you should our paper be accepted.

---

### Author Rebuttal · Authors · 2023-08-09

_Please see the attached PDF for figures cited in reviewer-specific rebuttals._

We thank all the reviewers for their helpful feedback. We would like to present two simple extensions of the theoretical results from our original submission, which we believe will make the paper even more compelling and useful for practitioners.

**Stronger theoretical result on surgicality** First, we prove a new theorem which implies that LEACE is "surgical" in a much stronger sense than we previously claimed. Specifically, we show that LEACE minimizes the mean squared distance between the original and edited features, not only with respect to the Euclidean norm, but with respect to _all_ norms of the form $\\| \mathbf x \\|_{\mathbf M}^2 = \mathbf x^T \mathbf{Mx}$ for some positive-semidefinite $\mathbf{M}$.

The proof is rather simple and intuitive. We begin with the case where $\mathrm X$ is centered and hence the LEACE bias term is zero. Without loss of generality, assume $\mathrm X$ is written in an orthogonal basis that diagonalizes $\mathbf M$. We can then write the reconstruction error as a weighted sum of $d$ terms, each of which only depends on a single row of the erasure matrix $\mathbf P$:

1. $\mathcal L(\mathbf P) = \mathbb E \Big [ \big\\| \mathbf P \mathrm X - \mathrm X \big\\|^2_{\mathbf M} \Big ] = \sum_{i=1}^d w_i \mathbb E \big [ (\mathbf P_i \mathrm X - \mathrm X_i)^2 \big ] = \sum_{i=1}^d w_i \mathcal L_i(\mathbf P_i),\quad \forall i : w_i \ge 0$.
The weights $w_i$ in Eq. 1 can be viewed as expressing the "importance" we assign to preventing changes along each component. For example, we might want to weight directions in proportion to their effect on the network's final output.

However, we can immediately see that these weights do not affect the set of optimal solutions. This is because Equation 1 implies that $\mathcal L$ is _additively separable_ along the rows of $\mathbf P$, and since our erasure constraint $\mathrm{Cov}(\mathbf P \Sigma_{\mathrm{XZ}}, \mathrm{Z}) = \mathbf 0_{d \times k}$ can also be decomposed into a set of independent constraints $\mathrm{Cov}(\mathbf P_i \Sigma_{\mathrm{XZ}}, \mathrm{Z}) = \mathbf 0_{1 \times k}$ for each row $\mathbf P_i$, we may conclude that $\mathbf P$ is optimal _if and only if_ each of its rows are optimal for their respective subproblems $\mathcal L_i$.

But this means that the optimality conditions for $\mathbf P$ are independent of the weights $w_i$, and hence also the choice of norm. Since we have already proven that the LEACE projection matrix is optimal for the Euclidean norm (i.e. $\mathbf M = \mathbf I$), we can conclude that it is optimal for all p.s.d. $\mathbf M$.

The extension to uncentered $\mathrm X$ closely mirrors the proof of Theorem 4.3 in our submission. Define $\tilde{\mathrm X} = \mathrm X - \mathbb E[\mathrm X]$ and $\mathbf{c} = \mathbf{P}\mathbb E[\mathrm X] + \mathbf{b} - \mathbb E[\mathrm X]$. Then we have \begin{align*} \mathbb E \big\\|\mathbf{P}\mathrm X + \mathbf{b} - \mathrm X \big\\|^2_\mathbf{M} &= \mathbb E \big\\|(\mathbf{P}\tilde{\mathrm X} - \tilde{\mathrm X}) + \mathbf{c} \big\\|^2_\mathbf{M} \\\\ & = \mathbb E \big\\|\mathbf{P}\tilde{\mathrm X} - \tilde{\mathrm X} \big\\|^2_\mathbf{M} + 2\mathbb E \big[ \mathbf{P}\tilde{\mathrm X} - \tilde{\mathrm X} \big]^T \mathbf{M} \mathbf{c} + \mathbf{c}^T \mathbf{M} \mathbf{c} \\\\ & = \mathbb E \big\\|\mathbf{P}\tilde{\mathrm X} - \tilde{\mathrm X} \big\\|^2_\mathbf{M} + \mathbf{c}^T \mathbf{M} \mathbf{c}, \end{align*} where we have eliminated the middle term because $\mathbf{P}$ is linear and $\mathbb E[\tilde{\mathrm X}] = 0$. Since $\mathbf{M}$ is p.s.d., our objective is minimized for $\mathbf{c} = \mathbf{0}$, i.e. $\mathbf{b} = \mathbb E[\mathrm X] - \mathbf{P}\mathbb E[\mathrm X]$. The problem thus reduces to choosing $\mathbf{P}$ so as to minimize $\mathbb E \big\\|\mathbf{P}\tilde{\mathrm X} - \tilde{\mathrm X} \big\\|^2_\mathbf{M}$ subject to $\mathrm{Cov}(\mathbf{P}\mathrm X + \mathbf{b}, \mathrm Z) = \mathrm{Cov}(\mathbf{P}\tilde{\mathrm X}, \mathrm Z) = \mathbf{0}$, which occurs when $\mathbf{P}$ is the LEACE projection matrix.

We are excited about this result because it suggests that LEACE should be useful under a wide variety of assumptions about which parts of a representation are most important. The choice of norm is not an arbitrary free parameter in our method.

**More intuitive closed-form formula & visualization** We also would like to present a new, yet _mathematically equivalent_ formula for the LEACE projection matrix which we believe is significantly more intuitive than the one we previously reported:

2. $r_{\mathrm{LEACE}}(\boldsymbol x) = \boldsymbol x - \mathbf{W}^+ \mathrm{Proj}(\mathbf{W}\Sigma_{\mathrm{XZ}}) \mathbf{W}\big (\boldsymbol x - \mathbb E[\mathrm X] \big )$,

where $\mathbf{W} = (\Sigma_{\mathrm{XX}})^{-1/2}$ is a whitening matrix and $\mathrm{Proj}(\mathbf{W}\Sigma_{\mathrm{XZ}})$ is the orthogonal projection matrix onto $\mathrm{colsp}(\mathbf{W}\Sigma_{\mathrm{XZ}})$.

Intuitively, Equation 2 tells us that LEACE de-means and whitens $\boldsymbol x$, projects onto the subspace responsible for correlations between $\mathrm X$ and $\mathrm Z$, then unwhitens the result. Finally, it subtracts this value from $\boldsymbol x$, thereby surgically removing the linearly available information about $\mathrm Z$. We encourage the reviewers to examine Figure 1 in the attached PDF, which shows this three-step process in action on a toy dataset.

We plan to include the full derivation of this new formulation in the appendix of the camera-ready version, where there is sufficient space. We welcome the reviewers to confirm numerically that this formula is indeed equivalent to the one we reported in our original submission, as long as $\Sigma_{\mathrm{XX}}$ is full rank.

---

### Decision · Program_Chairs · 2023-09-21

**Decision:**

Accept (poster)

**Comment:**

The reviewers have provided a comprehensive evaluation of the paper, highlighting its strengths and weaknesses. The paper's novel approach to concept erasure, its clear and simple proofs, and its compelling applications to language models have been appreciated. The paper's ability to improve upon previous projection approaches and provide a closed-form optimal solution to linear debiasing with the L2 regularization constraint is also commendable.

However, the reviewers have also pointed out areas for improvement, such as the need for more clarity on why the MSE reconstruction loss is a good measure of information being retained, the lack of comparison with other linear debiasing methods, and the need for more context about the debiasing setup for readers not too familiar with concept erasure literature.

Despite these concerns, the paper's strengths outweigh its weaknesses. The authors have made a significant contribution to the field of concept erasure, and their work is likely to have a high impact on at least one sub-area. Therefore, the paper is accepted. The authors are encouraged to address the reviewers' concerns in their final version of the paper.